# Progenitor Cells Activated by Platelet Lysate in Human Articular Cartilage as a Tool for Future Cartilage Engineering and Reparative Strategies

**DOI:** 10.3390/cells9041052

**Published:** 2020-04-23

**Authors:** Simonetta Carluccio, Daniela Martinelli, Maria Elisabetta Federica Palamà, Rui Cruz Pereira, Roberto Benelli, Ana Guijarro, Ranieri Cancedda, Chiara Gentili

**Affiliations:** 1Regenerative Medicine Laboratory, Department of Experimental Medicine (DIMES), University of Genova, via Leon Battista Alberti 2, 16132 Genova, Italy; simonetta.carluccio@edu.unige.it (S.C.); daniela.martinelli87@gmail.com (D.M.); elisabettapalama@gmail.com (M.E.F.P.); rui.pereira@iit.it (R.C.P.); ana.isabel.guijarro.anton@edu.unige.it (A.G.); 2Neurobiology of miRNA, Fondazione Istituto Italiano di Tecnologia, 16163 Genova, Italy; 3UOSD Oncologia Molecolare e Angiogenesi, IRCCS Ospedale Policlinico San Martino, largo Rosanna Benzi 10, 16132 Genova, Italy; roberto.benelli@hsanmartino.it; 4Endolife S.r.l., Piazza della Vittoria 15/23, 16121 Genova, Italy; ranieri.cancedda@unige.it; 5Center for Biomedical Research (CEBR), University of Genova, Viale Benedetto XV 9, 16132 Genova, Italy

**Keywords:** human articular cartilage, platelet lysate, chondro-progenitors, tissue regeneration

## Abstract

Regenerative strategies for human articular cartilage are still challenging despite the presence of resident progenitor cell population. Today, many efforts in the field of regenerative medicine focus on the use of platelet derivatives due to their ability to reactivate endogenous mechanisms supporting tissue repair. While their use in orthopedics continues, mechanisms of action and efficacy need further characterization. We describe that the platelet lysate (PL) is able to activate chondro-progenitor cells in a terminally differentiated cartilage tissue. Primary cultures of human articular chondrocytes (ACs) and cartilage explants were set up from donor hip joint biopsies and were treated in vitro with PL. PL recruited a chondro-progenitors (CPCs)-enriched population from ex vivo cartilage culture, that showed high proliferation rate, clonogenicity and nestin expression. CPCs were positive for in vitro tri-lineage differentiation and formed hyaline cartilage-like tissue in vivo without hypertrophic fate. Moreover, the secretory profile of CPCs was analyzed, together with their migratory capabilities. Some CPC-features were also induced in PL-treated ACs compared to fetal bovine serum (FBS)-control ACs. PL treatment of human articular cartilage activates a stem cell niche responsive to injury. These facts can improve the PL therapeutic efficacy in cartilage applications.

## 1. Introduction

Despite possessing resident progenitors committed to chondrogenesis, articular cartilage has poor intrinsic regenerative capability and minor cell turnover, mainly due to its avascular and alymphatic nature [1]. The isolation from the systemic circulation keeps away the inflammatory and reparative mechanisms triggered after injury, precluding damaged cartilage from recovering its functional configuration, which leads to tissue degeneration and the subsequent establishment of osteoarthritis (OA). Nowadays, OA is one of the major musculoskeletal diseases [2] and affects joints by causing gradual loss of articular cartilage together with osteophyte growth and synovial inflammation [3]. Since articular cartilage works as lubricant and load-bearing surface in healthy joints, symptoms of these disorders are pain and disability, which lead to an impaired patients’ quality of life and increased health-care costs for the society.

Currently, several therapeutic approaches for focal chondral defects and OA are used, including bone marrow surgical stimulation aimed at triggering intrinsic reparative mechanisms, cell therapy based on autologous chondrocyte implantation (ACI) and osteoarticular auto/allografts to fill and restore cartilage defects [4]. However, the treatment of extended defects is still challenging, especially for elderly people, and these techniques are still defective because the neo-tissue is often fibrotic and not completely integrated with neighboring tissues [5].

Therefore, research efforts are directed to explore new methods to achieve articular cartilage regeneration and repair. The use of therapies based on stem cells has attracted great interest since it is widely known that mesenchymal stem cells (MSCs), such as bone-marrow-derived MSCs (BM-MSCs), can differentiate into the chondrogenic lineage both in vitro and in vivo [6,7]. However, stem cell niches are located in situ, where they could participate directly in tissue homeostasis and repair processes. In fact, joint tissues, such as the synovium, hold MSCs with chondrogenic potential [8], and articular cartilage itself contains a postnatal progenitor cell population [9,10]. Moreover, during OA, pathological joint tissues and synovial fluid become enriched of cells with features of stemness [11,12], probably as an attempt to limit the ongoing damage and restore tissue integrity.

In the field of regenerative medicine, local transplantation or systemic infusion of stem cells represents an effective cellular therapy in many pathological states concerning the musculoskeletal system. Nevertheless, regeneration strategies targeting stem cells in situ could be more attractive and more advantageous thanks to the absence of in vitro culture steps and loss of injected cells. According to this approach, endogenous stem cells are recruited to the injury site by administration of bioactive factors. Thus, in the last decades, among a wide range of products, platelet-rich plasma (PRP) has spread as a clinical treatment tool for musculoskeletal diseases [13]. Since PRP or other platelet derivatives (i.e., platelet lysate (PL)) are a mix of growth factors, cytokines and chemokines normally involved in tissue healing, the rationale behind their application is the reactivation of latent endogenous regenerative mechanisms. Several studies have investigated PRP or PL roles both in vitro and in vivo, highlighting their capacity to exert anti-inflammatory and proliferating effects on cells [14,15,16], as well as to stimulate resident progenitors or to recruit circulating ones (together with immune and endothelial cells) [17,18].

Regarding cartilage disorders, treatments based on platelet-derived products have shown pain relief and functional improvement in patients, confirming their chondroprotective function in these pathologies [19,20,21]. From a mechanistic point of view, these beneficial outcomes could be explained by the fact that PRP-derived factors promote matrix deposition and downregulate inflammatory signalling in chondrocytes [22,23], and enhance migration and chondrogenic differentiation of progenitor cells [17] as well as cartilage tissue maturation [24].

However, the mechanism of action and efficacy of platelet derivatives in orthopaedics still need to be elucidated, especially due to the wide variety of preparation and standardization methods that can impact product composition [25], thereby affecting the physiological response. A better understanding of events leading to PRP- or PL-induced cartilage repair may allow solving these issues.

Here, we report that ex vivo treatment of human articular cartilage from hip joints with PL induces activation and outgrowth of cells that are endowed with some features of stemness, such as clonogenicity and expression of nestin [26], and higher proliferation capacity than resident chondrocytes with concurrent chondrogenic potential maintenance, both in vitro and in vivo. Stimulation of nestin-positive progenitor cells induced by PL in articular cartilage is of special interest for the future development of therapeutic strategies given the involvement of these cells in tissue regenerative processes. Moreover, we further characterize the PL effects on the phenotype of mature hip articular chondrocytes by showing that they reverted to an earlier stage similar to that of chondro-progenitor cells.

## 2. Materials and Methods

### 2.1. Platelet Lysate (PL) Preparation

Buffy coat samples obtained from the whole blood of healthy donors at the Blood Transfusion Center of the IRCCS Policlinico San Martino Hospital (Genova, Italy) were used to prepare PL. All the procedures were performed with the approval of the Institutional Ethics Committee and the Italian Ministry of Health: no. 423/2017-PR -7/7/2016, (D. lgs. 26/2014). According to Backly et al. [27], platelet pellet was obtained after serial centrifugation and resuspended at a concentration of 1 × 10^7^ platelets/µL in plasma to get PRP. Platelet membrane rupture in the PRP suspension was achieved by three cycles of immersion in liquid nitrogen for 1 min and incubation at 37 °C for 6 min. The suspension was centrifuged at 900 × *g* for 3 min at 4 °C and the supernatant was collected to obtain the PL, divided in aliquots and stored at −20 °C until use. Further details on platelet product standardization and safety were reported in [28,29].

In preliminary studies, several PL concentrations were tested (from 2.5 to 10%) on chondrocyte and cartilage cultures (data not shown). Five percent PL represents the maximum effective concentration in terms of cell responses (proliferation and outgrowth from tissue chips).

### 2.2. Cell Primary Cultures

#### 2.2.1. Chondro-Progenitor Cells (CPCs)

Human articular cartilage biopsies were harvested from patients (*N* = 20 with an age range from 31 to 88 years old, 65-year median age) undergoing hip replacement surgery. All tissue samples were obtained with written informed patients’ consent and according to the guidelines of the institutional Ethics Committee of IRCCS Policlinico San Martino Hospital (Genova, Italy), no. 423/2017-PR -7/7/2016. Articular cartilage was separated from subchondral bone and fragmented in slices, which were further cut into disks with a biopsy punch of 8 mm in diameter. Each disk was divided into two halves, and each half was then cultured in Dulbecco’s Modified Eagle’s Medium High Glucose (DMEM HG) containing 1 mM sodium pyruvate, 100 mM HEPES buffer, 1% penicillin/streptomycin and 1% L-glutamine (all from Euroclone, Milano, Italy) supplemented either with 10% fetal bovine serum (FBS) (Thermo Fisher Scientific, Waltham, MA, USA) or 5% PL in 6-well plates for 1 month (Figure 1A). Putative chondro-progenitor cells (CPCs), moving from cultured cartilage chip to the dish, were detached with trypsin/EDTA (Euroclone, Milano, Italy) and expanded in aforementioned medium supplemented with 5% PL (CPCs-PL).

#### 2.2.2. Primary Articular Chondrocytes (ACs)

Primary articular chondrocytes (ACs) were obtained as described by Pereira et al. [14] from the remaining cartilage biopsy (Figure 1A). In brief, chondrocytes were released by repeated digestions using an enzymatic solution composed of 1 mg/mL hyaluronidase (Sigma-Aldrich, St. Louis, MO, USA), 400 U/mL collagenase I, 1000 U/mL collagenase II (both from Worthington Biochemical, Lakewood, NJ, USA) and 0.25% trypsin (Thermo Fisher Scientific, Waltham, MA, USA). The cells obtained were plated in DMEM HG basal medium described above and containing 10% FBS. At ~90% of confluence, cells were trypsinized and split in culture medium supplemented with 10% FBS (ACs-FBS) or 5% PL (ACs-PL). During culture, cells were monitored using a bright-field microscope equipped with a digital camera (Leica DMi1; Leica Microsystems, Wetzlar, Germany). All described cell cultures were maintained in an incubator at 37 °C, with 5% CO_2_ and in normoxic oxygen condition.

### 2.3. Growth Kinetics

Growth kinetics were determined by plotting cell doublings of ACs-FBS, ACs-PL and CPCs-PL against time. Cell doublings were calculated considering the number of cells plated and recovered at each passage. Briefly, semi-confluent cells were trypsinized, counted and always replated at a density of 1.25 × 10^4^ cells/cm^2^ in 60 mm culture dishes. Six primary cultures were tested (*N* = 6).

### 2.4. Western Blot Analysis

At passage 2, confluent monolayers of ACs-FBS, ACs-PL and CPCs-PL were washed with phosphate-buffered saline 1X (PBS) and scraped in cold radioimmunoprecipitation assay (RIPA) buffer containing 50 mM Tris (pH 7.5), 150 mM sodium chloride, 1% deoxycholic acid, 1% triton X-100, 0.1% SDS, 0.2% sodium azide and proteinase inhibitor cocktail (Sigma-Aldrich, St. Louis, MO, USA). Protein extract concentration was quantified by Bradford assay (Serva Electrophoresis GmbH, Heidelberg, Germany) and Western blot was performed according to Nguyen et al. [30]. Equal amounts of total proteins (10 µg) were loaded on 4–12% NuPAGE Bis-Tris gel (Thermo Fisher Scientific, Waltham, MA, USA), and electrophoresis was performed. Gels were blotted onto nitrocellulose membranes (GE Healthcare Life Sciences, Uppsala, Sweden), immunoprobed overnight at 4 °C with primary antibodies raised against cyclin D1 (Abcam, Cambridge, UK) and α-tubulin (Sigma-Aldrich, St. Louis, MO, USA), both at a 1:10,000 dilution. After washing, membranes were exposed to horseradish peroxidase-linked goat anti-rabbit IgG at dilution of 1:5000 (GE Healthcare Life Sciences, Uppsala, Sweden) for 1 h at room temperature (RT), and bands were visualized using enhanced chemiluminescence (ECL, GE Healthcare Life Sciences, Uppsala, Sweden). Then, X-ray films (Fujifilm GmbH, Düsseldorf, Germany) were exposed to membranes, developed and fixed. Three primary cultures were tested (*N* = 3).

### 2.5. Evaluation of Cell Senescence

ACs-FBS, ACs-PL and CPCs-PL were analysed for senescence by detection of the senescence-associated β-galactosidase (SA-βgal) activity in a chromogenic assay (Sigma-Aldrich, St. Louis, MO, USA) according to previous protocol [31]. In brief, adherent cells in 24-well dishes were fixed in 3% paraformaldehyde (PFA) and stained overnight at 37 °C with fresh staining solution containing 40 mM citric acid/sodium phosphate buffer, 5 mM potassium ferrocyanide, 5 mM potassium ferricyanide, 150 mM sodium chloride, 2 mM magnesium chloride and 1 mg/mL 5-bromo-4-chloro-3-indolyl-d-galactoside (X-gal) in distilled water (all reagents from Sigma-Aldrich, St. Louis, MO, USA). Positive cells were observed under microscope Axiovert 200M (Carl Zeiss, Oberkochen, Germany) and counted in five regions of interest (ROI) at 20× magnification for each replicate. Stained cells were calculated as a percentage of the total number of cells on the plate. Cells from 5 donors were subjected to the assay at passage 2 (*N* = 5).

### 2.6. Assay for In Vitro and In Vivo Neoplastic Transformation of CPCs

To exclude malignant properties of CPCs, in vitro colony assay formation and in vivo tumorigenesis were investigated. For in vitro test, anchorage-independent growth assay in methylcellulose media was conducted [32]. Thus, CPCs-PL at passage 3 were plated at a density of 10,000 cells/35 mm petri dishes in the semi-solid culture system provided by StemMACS HSC-CFU Media (Miltenyi Biotec, Bergisch Gladbach, Germany) according to the manufacturer’s protocol. After 14 days of incubation, formation of colony-forming unit (CFU)-cells (CFU-C) was assessed under an inverted microscope (Leica DMi1; Leica Microsystems, Wetzlar, Germany). Three primary cultures were tested (*N* = 3). MDA-MB-231 triple-negative breast cancer cell line was used as positive control. In vivo tumorigenesis was assessed with CPCs-PL at passage 2 from two different pools. For each pool, 1 × 10^6^ cells were injected subcutaneously into NOD/SCID mice (*N* = 12 animals, males and females, 6–8 weeks old). The mice were monitored up to 2–3 months. All experimental animal procedures were evaluated and approved by Ethics Committee for animal experimentation (CSEA) and communicated to the Italian Ministry of Health in accordance with article 31 of the D. lgs 26/2014 (approval no. 29/2017-PR, 16/01/2017 by Italian Ministry of Health). 

### 2.7. RNA Extraction and Real-Time Quantitative Reverse Transcription Polymerase Chain Reaction (qRT-PCR)

Total RNA from ACs-FBS, ACs-PL and CPCs-PL at passage 1 grown upon confluence (in 100 mm dishes) was extracted by TRIzol™ Reagent (Thermo Fisher Scientific, Waltham, MA, USA) according to the manufacturer’s protocol. RNA concentrations were measured at 260 nm using Nanodrop TM 1000 (Thermo Fisher Scientific, Waltham, MA, USA) and RNA purity was checked considering 260 nm/280 nm ratio with values included in the 1.5–2.1 range. Complementary DNA (cDNA) synthesis was performed starting from 1 μg of total RNA and using SuperScript First-Strand synthesis system for reverse transcription polymerase chain reaction (RT-PCR) (Thermo Fisher Scientific, Waltham, MA, USA) following the manufacturer’s instruction. Transcript levels of target genes were measured by real-time quantitative PCR (qRT-PCR) using Power SYBR^®^ Green PCR Master Mix on 7500 Fast Real-Time PCR System (Applied Biosystems, Foster City, CA, USA). The housekeeping gene *GAPDH*, commonly utilized as a reference gene in cartilage and bone qPCR assays [33], was used as the endogenous control for normalization. The selected human-specific primer sequences were as follows: type II collagen (*COL2A1*), forward 5′ - GGCAATAGCAGGTTCACGTACA - 3′, reverse 5′ - CGATAACAGTCTTGCCCCACTT - 3′; type I collagen (*COL1A1*), forward 5′ - CAGCCGCTTCACCTACAGC - 3′, reverse 5′ - TTTTGTATTCAATCACTGTCTTGCC - 3′; *SOX9*, forward 5′ - CCCGCACTTGCACAACG - 3′; reverse 5′ - TCCACGAAGGGCCGCT - 3′; nestin (*NES*), forward 5′ - CAGAGGTGGGAAGATACGGT - 3′, reverse 5′ - AGCTCTGCCTCATCCTCATT - 3′; *GAPDH*, forward 5′ - CCATCTTCCAGGAGCGAGAT - 3′, reverse 5′ - CTGCTTCACCACCTTCTTGAT - 3′. Data were reported as ratio *GENE*/*GADPH*, which represents the expression of the gene of interest divided by the expression of *GAPDH* in the same sample.

### 2.8. Immunofluorescence Staining and Immunophenotypic Characterization by Flow Cytometry

To perform immunofluorescence staining, ACs-FBS, ACs-PL and CPCs-PL at passage 1 were seeded on coverslips at a density of 10^5^ cells/cm^2^ and fixed with 3.7% PFA after 3–4 days of culture. Fixed cells were permeabilized with a solution containing 20 mM HEPES (pH 7.4), 300 mM sucrose, 50 mM sodium chloride, 3 mM magnesium chloride and 0.5% triton X-100. After blocking with 20% normal goat serum (NGS, Thermo Fisher Scientific, Waltham, MA, USA), samples were incubated overnight at 4 °C with primary antibodies raised against SOX9, 1:200 diluted in 10% NGS (Abcam, Cambridge, UK); type II collagen, 1:250 diluted in 10% NGS (CIICI-Developmental Studies Hybridoma Bank, University of Iowa); type I collagen, 1:300 diluted in 10% NGS (SP1.D8-Developmental Studies Hybridoma Bank, University of Iowa); nestin, 1:2000 diluted in 10% NGS (Abcam, Cambridge, UK); and cleaved caspase-3, 1:400 diluted in 10% NGS (Cell Signaling Technology, Danvers, MA, USA). Positive staining was detected by incubation with Alexa Fluor 488- or Alexa Fluor 594-conjugated anti-mouse or anti-rabbit immunoglobulin IgG secondary antibodies diluted 1:300 in NGS 10% (Thermo Fisher Scientific, Waltham, MA, USA) for 1 h at RT, followed by nuclear labelling with DAPI (Sigma-Aldrich, St. Louis, MO, USA). Negative control staining was performed without primary antibody incubation. Samples were observed under epifluorescent illumination using an Axiovert 200M microscope, and images were captured with AxiocamHR camera (Carl Zeiss, Oberkochen, Germany).

Expanded ACs-FBS, ACs-PL and CPCs-PL at passage 2 were phenotypically characterized for a set of surface markers using flow cytometry. After trypsinization, 1 × 10^5^ cells were incubated separately with 1 µL of one of the following fluorescein isothiocyanate (FITC)- or phycoerythrine (PE)-conjugated antibodies: CD44-FITC, CD166-PE, HLA-ABC-PE, HLA-DR-FITC (all from BD Pharmigen), CD90-PE, CD105-PE, CD73-FITC, CD146-FITC, CD106-PE, CD45-FITC, CD34-PE, CD29-PE and isotype-matched IgG-PE and IgG-FITC control antibodies (all from eBiosciences/ ThermoFisher Scientific, Waltham, MA, USA). The staining was performed for 30 min at 4 °C in the dark to preserve the fluorochromes. Samples were run on a CyAN ADP cytofluorimeter (Beckman-Coulter, Brea, CA, USA). Data were analyzed using FlowJo V10 software (Tree Star Inc., San Carlos, CA, USA) and expressed as Log fluorescence intensity versus number of cells. Experiments were repeated on three different primary cultures (*N* = 3).

### 2.9. Colony-Forming Unit Fibroblast (CFU-F) Assay

Clonogenic potential of ACs-FBS, ACs-PL and CPCs-PL at passage 1 was explored by plating them at low density (10 cells/cm^2^) in 100 mm culture dishes and performing the colony staining after 12 days of culture. At the end of the culture time, cells were washed with PBS, fixed with 3.7% PFA in PBS for 15 min at RT and stained with 1% methylene blue in borate buffer (10 nM, pH 8.8) for 45 min at RT. CFU-F assay was performed in duplicate for each tested primary culture (*N* = 10). Colony-forming efficiency (CFE%) was calculated as follows: (number of colonies formed/number of plated cells) × 100. A set of 6-well dishes were prepared for detection of nestin by immunofluorescence.

### 2.10. In Vitro Multilineage Differentiation Potential

The chondrogenic potential of CPCs grown in 5% PL and ACs expanded in the presence of either FBS or PL was checked at passage 1 by micromass pellet culture in vitro (*N* = 3). About 2.5 × 10^5^ cells were pelleted in conical tubes and cultured for 3 weeks in chondrogenic medium containing 10 ng/mL human transforming growth factor-β1 (hTGF-β1) (PeproTech, Rocky Hill, NJ, USA), 10^−7^ mol/L dexamethasone and 50 mg/mL ascorbic acid (both from Sigma-Aldrich, St. Louis, MO, USA) according to Johnstone et al. [6]. Chondrogenic differentiation was subsequently investigated by histological staining with toluidine blue (see Section 2.12. below).

To test osteogenic differentiation, cells were seeded in 24-well plates at a density of 10^5^ cells/cm^2^ in the presence of osteogenic induction medium containing 5 μg/mL ascorbic acid, 10^−7^ mol/L dexamethasone and 10 mmol/L β-glycerophosphate (all from Sigma-Aldrich, St. Louis, MO, USA). After 3 weeks of culture, calcium deposits were stained by Alizarin Red S (Sigma-Aldrich, St. Louis, MO, USA) solution.

To induce adipogenesis, cells were seeded as just reported above and grown in culture medium containing 1 μmol/L dexamethasone, 60 μmol/L indomethacin, 10 μg/mL insulin and 1 mmol/L 3-Isobutyl-1-methylxanthine (IBMX) (all from Sigma-Aldrich, St. Louis, MO, USA). After 3 weeks of culture, intracellular lipid drops were detected with Oil Red O staining (Sigma-Aldrich, St. Louis, MO, USA). Osteogeninc and adipogenic potential of CPCs and ACs were determined at passage 2 in triplicate on three different primary cell cultures (*N* = 3).

### 2.11. In Vivo Cartilage and Bone Formation

CPCs-PL chondrogenic and osteogenic potential in vivo was investigated by implantation of cell pellets and cell-seeded biomaterials in athymic mice (female CD-1 nu/nu, 6–8 weeks old; Charles River Laboratories Italia, Lecco, Italy). CPC pellets were obtained as already described in Section 2.10 and implanted subcutaneously in mice after three days of in vitro culture in chondrogenic medium.

Moreover, CPCs-PL at passage 2 were also seeded on absorbable polyglycolic-acid–hyaluronan (PGA-HA) scaffolds (BioTissue AG, Zurich, Switzerland) to detect cartilage formation and/or on calcium phosphate ceramic scaffolds (hydroxyapatite/β-tricalcium phosphate, HA/β-TCP, Biomatlante, Vigneux de Bretagne, France) for osteogenic induction. Briefly, CPCs-PL were trypsinized at passage 1 and 2 × 10^6^ cells were resuspended in 33% *v/v* fibrinogen in PBS (Tissucol, Baxter Healthcare, Deerfield, IL, USA). Both types of scaffolds were soaked with CPCs suspension, and fibrinogen was polymerized by the addition of 1:10 *v/v* thrombin in PBS (Tissucol, Baxter, Healthcare, Deerfield, IL, USA). Constructs with ACs-FBS at passage 1 associated with ceramic granules were also prepared as control. Cell grafts were maintained in chondrogenic or osteogenic medium for 3 days before subcutaneous implantation in mice. A number of at least three primary cultures was used for these experiments. Groups of 8 animals were sacrificed 4 and 8 weeks after surgery for chondrogenesis or osteogenesis, respectively, and the harvested implants were processed for the histological analysis to evaluate cartilage and bone formation. All experimental animal procedures were evaluated and approved by Ethics Committee for animal experimentation (CSEA) and communicated to the Italian Ministry of Health in accordance with article 31 of the D. lgs 26/2014 (approval no. 29/2017-PR, 16/01/2017 by Italian Ministry of Health). 

### 2.12. Histology and Immunohistochemistry

Cartilage chips, pellet and implant samples were fixed in 3.7% PFA in PBS, dehydrated in ethanol, and paraffin-embedded. Cross sections of 5 µm were cut (by using microtome RM2165, Leica Microsystems, Wetzlar, Germany), dewaxed and stained according to the appropriate histological analysis: haematoxylin and eosin staining to observe cell organization and toluidine blue staining to detect sulphated glycosaminoglycans in cartilage. For immunohistochemical analysis, dewaxed sections were treated with methanol/hydrogen peroxide (49:1) solution for 30 min to inhibit endogenous peroxidase activity, then permeabilized with 0.3% triton X-100 in PBS for 10 min and finally incubated with hyaluronidase (Sigma-Aldrich, St. Louis, MO, USA) at a concentration of 1 mg/mL in PBS (pH 6.0) for 30 min at 37 °C. After washes in PBS and incubation with 20% NGS for 1 h to inhibit nonspecific binding, the slices were incubated overnight at 4 °C with primary antibodies raised against: type II collagen, 1:250 diluted in 10% NGS (CIICI-Developmental Studies Hybridoma Bank, University of Iowa); type X collagen, 1:1000 diluted in 10% NGS (Abcam, Cambridge, UK); and proliferating cell nuclear antigen (PCNA), 1:200 diluted in 10% NGS (Abnova, Taipei City, Taiwan). The immunobinding was detected by incubation with biotinylated secondary anti-mouse or anti-rabbit antibodies (Dako, Agilent Technologies Inc., Carpinteria, CA, USA) for 30 min at RT followed by treatment with streptavidin-peroxidase (Jackson ImmunoResearch Laboratories Inc., West Grove, PE, USA). Peroxidase activity was finally visualized by 3-amino-9-ethylcarbazole (Sigma-Aldrich, St. Louis, MO, USA) chromogen substrate. Images were acquired by a microscope Axiovert 200M (Carl Zeiss, Oberkochen, Germany) at different magnifications. Data for PCNA quantification were summarized in a histogram reporting the PCNA positive cell percentage per field view for cartilage chips cultivated in 10% FBS or 5% PL (*N* = 3).

### 2.13. Production of Cell-Conditioned Media

ACs-FBS, ACs-PL and CPCs-PL at passage 1 were grown until 80% of confluence, extensively washed with PBS and incubated with DMEM HG culture medium without any supplements for 24 h of conditioning. Conditioned media (CM) from each condition (ACs 10% FBS-CM, ACs 5% PL-CM and CPCs 5% PL-CM) were collected and centrifuged at 300 × *g* for 10 min and then at 2000 × *g* for 20 min, and supernatants were stored in aliquots at −80 °C. In cytokine array experiment, supernatants in each condition were further concentrated by using Amicon™ Ultra Centrifugal Filter Units with 3KDa molecular weight cut-off (Merck Millipore, Burlington, MA, USA). Amount of proteins in CM from CPCs and ACs was quantified by performing Bradford assay (Serva Electrophoresis GmbH, Heidelberg, Germany).

### 2.14. Cytokine Identification in Cell Secretome

The release of cytokines and chemokines in CPCs- and ACs-CM was analysed using the Human XL Proteome Profiler™ Array (R&D Systems, Minneapolis, MN, USA) according to the user’s manual. Briefly, membranes spotted with antibodies were incubated with the same amounts (50 µg/mL) of each CM overnight at 4 °C. The following day, detection antibody cocktail was added for 1 h at RT, before visualization using ECL. Quantitative analysis was performed on scanned (Epson perfection 1260 scanner, Seiko Epson Corporation, Nagano, Japan) X-ray films (Fujifilm GmbH, Düsseldorf, Germany) using the Protein Array Analyser plugin available for ImageJ software (US National Institutes of Health, Bethesda, MD, USA). For each membrane, average spot signal density was determined by densitometry, followed by background subtraction and normalization to the reference spots. For a correspondence between a specific molecule and its position in the membrane, a reference to the manufacturer’s datasheet is provided (Catalog # ARY022B).

### 2.15. In Vitro Chemotaxis of CPCs

CPCs-PL migration was investigated by Boyden chamber assay using serum-free medium as negative control and ACs-CM pre-treated for 24 h with IL-1β (PeproTech, Rocky Hill, NJ, USA), 5% PL or both stimuli as chemoattractants (see Section 2.13 for CM preparation). Cells were plated at a density of 12 × 10^4^/chamber on the top of the filter inserts and incubated for 4 h at 37 °C with 5% CO_2_. Cells migrated to the lower surface of the filters were fixed in ethanol, stained with toluidine blue and quantified by a bright field microscope (Leica DMi1; Leica Microsystems, Wetzlar, Germany). Data were reported as fold-increase migration, which represented migration extent of CPCs in each condition referred to the control (migration under the influence of conditioned media from non-treated ACs) set to 1 value. Each experiment was performed in triplicate and repeated at least three times (*N* = 3).

### 2.16. In Vitro Scratch Assay

ACs at passage 1 were plated in 6-well plates, cultured until confluence and treated with either 10% FBS or 5% PL for 24 h. In parallel, CPCs-PL were cultured until confluence. Cell monolayers were washed extensively with PBS to remove residual of factors, scratched using 100 µL pipette tips and covered with serum-free DMEM HG culture medium. Scratch closure was monitored from t_0_ = 0 h to t_1_ = 24 h and t_2_ = 48 h with inverted microscope (Leica DMi1, Leica Microsystems, Wetzlar, Germany). Analysis of acquired images was performed with TScratch software (https://github.com/cselab/TScratch) as reported by Romaldini et al. [15]. Experiments were performed in triplicate on three different primary cultures (*N* = 3).

### 2.17. Statistical Analysis

All data are presented as means and standard deviation (SD) or standard error of the mean (SEM) when based on the mean of duplicates or triplicates. Normal distribution of values was assessed by the Shapiro–Wilk normality test. Unpaired Student’s *t*-test was used to determine statistical significance for normally distributed data and Mann–Whitney test in the absence of a normal distribution to compare ACs-PL versus ACs-FBS or ACs-PL versus CPCs-PL. Level of significance was set at *p* < 0.05 (* *p* < 0.05, ** *p* < 0.01, *** *p* < 0.001, **** *p* < 0.0001). Kruskal-Wallis with Dunn’s multiple comparison test was used to analyse in vitro scratch assay and one-way ANOVA for in vitro chemotaxis assay followed by post-hoc Sidak’s multiple comparisons test for group differences. Data were analyzed with GraphPad Prism^®^ 8.0 software (GraphPad Software, Inc., San Diego, CA, USA).

## 3. Results

### 3.1. PL Induced Release of Cells with Fibroblastic-Like Phenotype from Ex Vivo Cultured Cartilage Chips and Promoted Their Proliferation

Articular cartilage biopsies from patients were divided in fragments, some of which were enzymatically digested to obtain primary cultures of ACs, and others were directly cultured in vitro in petri dishes. Each fragment used for organ culture was divided into two halves. One of them was cultured in 10% FBS and the other one in 5% PL (Figure 1A). This approach allows to consider the observed tissue reactions as a consequence of the culture conditions and not due to sample heterogeneity.

After 15–20 days of culture in the presence of PL, spindle-shaped putative CPCs derived from tissue and attached to the bottom of the plate could be observed (Figure 1B) as opposed to FBS-control culture, in which, in most cases, no cells exited from the cartilage fragments. In the rare cases in which cells were derived from tissue cultured in FBS, those cells were unable to proliferate or to be expanded for further analysis. Immunohistochemistry performed on cartilage chips after cell harvesting showed higher positivity to the proliferation marker PCNA in cartilage slices treated with PL compared to FBS ones (56.9 ± 5.7 and 17.8 ± 8.9, respectively with *p* < 0.05) (Figure 1C,D).

Previous studies reported that CPC frequency and proliferation increase in pathological cartilage [34,35], but they are not influenced by donor age [34,36].

### 3.2. PL Increased the Proliferation of ACs and Reduced Their Senescence

Growth kinetics analysis (Figure 2A) showed a similar proliferative rate of CPCs-PL and ACs-PL throughout the culture time, while ACs-PL displayed higher cell doublings than ACs-FBS at all the time points analysed (9.4 ± 1.1, 10.2 ± 2.6 and 1.3 ± 0.7 cell doublings at 23 days in culture, respectively). The assessment of cyclin D1 levels, a marker of cell cycle progression [37], in confluent cells by Western blot confirmed the higher proliferative rate of ACs expanded in PL versus almost quiescent ACs grown in FBS as well as the similar growth of CPCs-PL compared to ACs-PL (Figure 2B). These experimental data highlight the ability of platelet-derived products to induce a strong mitogenic response on cells, especially on chondrocytes, which usually have a very low turnover.

Given the variation observed in cell growth kinetics, we investigated cell senescence by detecting the SA-βgal activity in a chromogenic assay. This test performed on enzymatically isolated ACs showed that the percentage of βgal-positive cells significantly decreases in the PL-treated group compared to FBS group (20 ± 5 and 39 ± 4, respectively with *p* < 0.05). CPC population recruited from cartilage chips by PL contains a comparable fraction of labelled cells with respect to ACs grown in PL (12 ± 2 and 20 ± 5, respectively, with no statistically significant difference). Representative images of the three experimental cell groups after staining and the percentage of positive cells are shown in Figure 2C. Among the beneficial effects of platelet derivatives, chondroprotection could probably also occur by reducing the apoptosis rate and a recent study seems to confirm this tendency [38]. A preliminarily study on apoptosis in ACs and CPCs was performed by the detection of cleaved caspase-3, which represents a reliable marker for cells that are dying by apoptosis [39] in immunofluorescence experiments (Appendix A). Contrary to Moussa et al. [38], we did not detect any evident difference in cleaved caspase-3 expression between the three examined experimental groups; rather, its levels appeared quite low. Furthermore, it could be necessary to induce stress or apoptotic signals in order to trigger and subsequently detect an appreciable anti-apoptotic effect of PL.

To exclude any risk in view of clinical applications, tumorigenic potential of CPCs was tested in vitro by anchorage-independent growth in methylcellulose media [32]. No CFU-C were detected in cultured CPCs, contrary to positive control metastatic breast cancer cell line MDA-MB-231, as expected (Figure 2D). Moreover, the safety of CPCs was also confirmed in vivo by cell implantation in NOD/SCID mice and by monitoring the animals up to 12 weeks. No mouse developed tumours or macroscopic nodules and showed signs of illness. Finally, autopsy did not reveal anatomical abnormalities.

### 3.3. Effect of PL on Gene Expression and Phenotype in Cartilage-Derived Cells

Since ACs and CPCs share the same anatomical location and the former can dedifferentiate and acquire stemness-like characteristics in culture [40], the distinction between the two populations is difficult. In this context, we analyzed phenotype of CPCs-PL in parallel with that of ACs-PL. Moreover, we compared ACs-PL versus ACs-FBS. The analysis of typical chondrocyte markers by qRT-PCR (Figure 3A) showed a drastic reduction of type II collagen (*COL2A1*) in ACs-PL compared to ACs-FBS (*p* < 0.05). Statistical differences were also observed between ACs-PL and CPCs-PL (*p* < 0.01) in the expression of this marker. The expression of the master regulator *SOX9* was also decreased in ACs-PL compared to ACs-FBS (*p* < 0.05), while it was comparable in ACs-PL and CPCs-PL. Levels of type I collagen (*COL1A1*) were instead not significantly different among all analyzed cell types. However, these differences in the chondrogenic gene levels were less marked at protein level among the examined groups, as shown in immunofluorescence experiments (Figure 3B). Flow cytometric analysis performed in the three cell populations reported almost complete positivity for a common set of surface markers such as CD90, CD73, CD105, CD44, CD29 and HLA-ABC (HLA class I), typically expressed by mesenchymal stem/progenitor cells and chondrocytes after in vitro expansion [41]. Hematopoietic markers (CD45 and CD34) and HLA-DR (HLA class II) were not expressed, and CD146, a member of the immunoglobulin (Ig) superfamily of cell adhesion molecules (CAMs), was negative in ACs-FBS and ACs-PL and, although a low positivity in one case, also in CPCs-PL. Differences in the percentages of CD106^+^ and CD166^+^ cells were observed among the three cell populations. The levels of positivity for CD106, constitutively expressed by human articular chondrocytes [42], did not significantly change between ACs-FBS and ACs-PL cultured in vitro (59 ± 9% and 41 ± 8%, respectively), while the percentage of CD106^+^ cells was lower in CPCs-PL than in ACs-PL (16 ± 5% and 41 ± 8%, respectively, with *p* < 0.05). Conversely, ACs-PL were enriched of CD166^+^ cells compared to ACs-FBS (59 ± 10% and 29 ± 8%, respectively, with *p* < 0.05) and CPCs-PL displayed not statistically different levels of CD166 expression versus ACs-PL (72 ± 8% and 59 ± 10%, respectively).

Previous work reported a cell population in human adult articular cartilage that co-expresses CD105 and CD166 as multi-potent mesenchymal progenitors [12]. A representative experiment and relative percentages of positive cells are reported in Figure 3C,D, respectively.

### 3.4. PL Modulated the Clonogenic Potential and Expression of Nestin Stem Marker in Cartilage-Derived Cells

Colony-forming ability is a recognised trait of stem/progenitor cells. CPCs obtained from cartilage chips cultured in PL were able to form colonies at low-density plating (Figure 4A). When we tested the CFU-F potency using mature ACs cultured in FBS and PL conditions, interestingly, we observed that ACs-PL were able to recover this potential, which was absent in ACs-FBS (sporadic cell spots). Colony-forming efficiency was comparable between CPCs and ACs-PL (with no statistically significant difference and colony counts of 78 ± 11 and 81 ± 15, respectively), implying a probable PL involvement in this cell potentiality (Figure 4B). Moreover, CPCs and ACs were investigated for nestin expression by qRT-PCR and immunofluorescence experiments. Nestin is an intermediate filament protein involved in cytoskeleton remodelling in several tissues [43]. Nestin has been often considered as a progenitor cells marker, and it has been reported that its levels are downregulated during cell differentiation [44]. This evidence supports the presence of nestin in self-renewing cells that formed colonies in CFU-F assay among CPCs and ACs populations exposed to PL (Figure 4C). As shown in Figure 4E, CPCs-PL expressed nestin at higher level than ACs-PL (*p* < 0.05) during culture expansion. In mature chondrocytes, the level of nestin (*NES)* was low, although there was a slight but not statistically significant increase in ACs-PL compared to ACs-FBS at least on passage 1. These data were confirmed by immunofluorescence staining experiments (Figure 4D) where we detected no nestin-positive cells in ACs-FBS, scant stained cells among ACs-PL and a more constant presence of nestin-expressing cells in CPCs-PL.

### 3.5. Comparison of In Vitro Multilineage Differentiation Potential Between CPCs and ACs

CPCs-PL were tested in vitro for their chondro-, osteo- and adipogenic potential in parallel with ACs in both culture conditions (PL and FBS) as shown in Figure 5. Pellet culture is suitable to investigate chondrogenic differentiation in vitro [6]. Thus, after 21 days of induction in pellet culture, both CPCs and ACs were able to produce metachromatic matrix, as shown by toluidine blue staining. The size of the recovered pellets was different among the examined groups. In particular, CPCs-PL pellets were bigger than the ACs-PL ones and, interestingly, showed a tissue-like organization. In turn, ACs-PL pellets were bigger than ACs-FBS ones (Figure 5A). Osteogenic differentiation was estimated based on calcium deposition in culture after 21 days of induction in vitro. CPCs-PL showed positivity to alizarin red staining as well as ACs (Figure 5B). Similarly, when cells underwent adipogenesis induction, clusters of oil red O positive lipid vacuoles were observed in all three populations (Figure 5C).

Since multi-lineage potential is one of the features hold by cells endowed with stemness properties, these data showed that adult articular cartilage contains cells able to give trilineage differentiation as MSCs. PL treatment may act on cartilage in order to select and mobilize these progenitor cells.

### 3.6. CPCs-PL Produced Hyaline-Like Cartilage In Vivo Suitable for Tissue Engineering Strategies

Chondrogenic potential of CPCs-PL was investigated by ectopic implantation of pellets and cell-seeded biomaterials in immunodeficient mice. Both types of implants led to a well-defined cartilage tissue-like formation: 1 month after implantation, CPC-pellets (Figure 6A) and CPCs-seeded constructs of PGA-HA biomaterials (Figure 6B) showed metachromatic extracellular matrix—and therefore are rich in proteoglycans—and positivity to type II collagen. Weak or absent staining for type X collagen, a marker of hypertrophy, was found in the histological analysis. After these results obtained in vitro, the osteogenic potential of CPCs was assessed in vivo. CPCs, combined with the osteoinductive ceramic granules, as described by Pereira et al. [45], formed a compact fibrous tissue, virtually ascribable to an immature bone-like matrix, 2 months after implantation in mice (Figure 6C). Altogether, these data demonstrated that CPCs were preferentially committed to a stable chondrogenic fate, a feature that is crucial for cartilage regeneration purposes and make CPCs a better option than traditional MSCs, usually predetermined to develop from transient endochondral cartilage to bone [46].

### 3.7. Secretory Profile of CPCs and ACs Revealed an Intricate Scenario Including Tissue Turnover, Hypertrophy Counteraction, PL-Induced Pro-Inflammatory Effects and Chemoattractive Capability

Conditioned media (CM) collected from cultures of ACs-FBS, ACs-PL and CPCs-PL were characterized by using cytokine array (Figure 7A, Appendix A). Detection of angiogenic molecules (angiogenin and vascular endothelial growth factor (VEGF)), molecules expressed in deep and hypertrophic cartilage layers (osteopontin, OPN) and other factors usually related to several disorders (B cell activating factor, BAFF/BLys) in the three experimental groups could be explained by the degenerative state of tissue biopsies harvested from patients undergoing hip replacement.

Molecules usually secreted from ACs were identified in analyzed CM, including chitinase-3-like protein 1 (CHI3L1), insulin-like growth-factor-binding protein-2 and -3 (IGFBP-2, -3), serpin E1, cystatin C (CST3), apolipoprotein A1 (ApoA1), thrombospondin 1 (THBS1) and pentraxin 3 (PTX3), as previously reported [47].

Some general observations could be made about differences in catabolic activity among the analyzed experimental groups. CPCs-PL secreted dipeptidyl peptidase-4 (DPPIV/CD26), not detected in ACs-PL, and a higher amount of the cysteine protease inhibitor cystatin C than ACs-PL. Levels of these two factors were comparable between ACs-FBS and ACs-PL. The serine proteinase inhibitor Serpin E1, also known as plasminogen activator inhibitor 1, was instead less released by ACs-PL, than by ACs-FBS, while no differences were observed between ACs-PL and CPCs-PL. uPAR, urokinase-type plasminogen activator receptor [48], increased in ACs-PL in comparison with ACs-FBS.

The results also offered some interesting cues concerning the possible modulation of cell hypertrophic fate. In this context, Dkk-1, an antagonist of the Wnt/β-catenin signalling pathway that prevents hypertrophy and cartilage degradation [49], was not released by ACs-FBS but slightly produced by ACs-PL. CM from CPCs-PL displayed markedly higher levels of this factor in comparison with CM from ACs-PL. On the contrary, in comparison with ACs-PL, CPCs-PL released a very low amount of Interleukin 1 receptor-like 1 (ST2/IL1RL1), which has recently been described as a RUNX2 target, and it is expressed during hypertrophic differentiation in chondrocytes [50]. Moreover, the amount of OPN released by CPCs-PL was much lower than that released by ACs-PL. Retinol binding protein 4 (RBP4) markedly decreased in CPCs-PL secretome compared to ACs-PL. RBP4 expression has been reported in epiphyseal chondrocytes of secondary ossification regions and a role in OA pathogenesis has been suggested in the literature [51,52].

Finally, pro-inflammatory cytokines, i.e., interleukin-8 (IL-8), interleukin-6 (IL-6) and lipocalin-2 (LP2/NGAL), and chemokines, i.e., monocyte chemotactic protein 1 (MCP-1/CCL2) and -3 (MCP-3/CCL7), were found as secreted molecules by all experimental cell groups. In particular, the first three were upregulated in CM of ACs-PL in comparison with ACs-FBS. In fact, the pro-inflammatory effect of PL is widely documented in triggering cascades of pro-resolving events [14,15,27] that could lead to regeneration in injured sites, just like it happens physiologically during wound healing. IL-6 and NGAL released by CPCs-PL was lower than that released by ACs-PL, while the release of IL-8 was similar in both cell populations.

Other soluble chemokines, i.e., CXCL1 (GROα), CXCL4, CXCL5, CXCL12 (SDF1), CCL5 (RANTES) and CCL20 were observed in the secretome of ACs-PL but not detected in the ACs-FBS-derived one. Except for CXCL1 and CCL5 levels, which were comparable, CXCL5, CXCL4 and CCL20 decreased markedly in CPCs-PL, while CXCL12 increased in comparison with ACs-PL. These molecules are usually involved in the regulation of leukocyte trafficking and stem cell chemoattraction [53,54], but it has also been suggested that these chemokines could have a role in cartilage physiological turnover [55].

### 3.8. PL-Recruited CPCs Showed Enhanced Motility Under Inflammatory Conditions

Since our final aim was to support a biological basis for possible therapeutic application of PL on cartilage defects, migratory capability of ACs and CPCs was tested by an in vitro scratch assay (Figure 7B). Cell motility was evident in experimental groups between 24 and 48 h as shown by wound healing assay (ACs-FBS: from 18.9 ± 9.5% to 29.6 ± 15.4% with no statistically significant difference; ACs-PL: from 36 ± 27.9% to 62 ± 30.6% with *p* < 0.05; CPCs-PL: from 46.7 ± 19.5% to 95.1 ± 5.3% with *p* < 0.0001). Interestingly, PL treatment significantly promoted ACs motility in the migration assay compared to ACs cultured in FBS. The scratch width in ACs-PL was significantly reduced at 48 h (*p* < 0.05) in comparison to ACs-FBS. As expected, CPCs were also able to migrate and close the scratch during the assay, to a similar extent within 24 h but with a statistically significant increase at 48 h compared to ACs-PL (*p* < 0.01). Thus, the marked migratory behaviour of CPCs was further tested by mimicking the inflammatory environment of a pathological state, i.e., by exposition to the CM of ACs that underwent treatment with the inflammatory cytokine IL1-β concurrently or not with PL exposition (Figure 7C). Under basal conditions, CM of ACs-PL was not able to act as a chemoattractant for CPCs when compared with the effects of CM from ACs-FBS. However, an enhanced chemotaxis on CPCs occurred in an inflammatory milieu, i.e., in presence of CM both derived from IL1-β-treated ACs (*p* < 0.05) and PL plus IL1β-treated ACs (*p* < 0.01), compared to control (ACs-FBS CM).

Altogether, these experiments showed that cartilage cells increase motility after PL treatment, despite the fact that they do not normally show this behaviour in cartilage. This effect probably allowed the escape of CPCs from cartilage chips. Furthermore, CPCs showed a strong reactivity to inflammatory stimuli. Due to these properties, CPCs and ACs activated by PL may be recruited into injured sites in vivo, contributing to the reparative process.

## 4. Discussion

Given the therapeutic use of platelet derivatives in orthopaedics, the aim of our study was to explore their stimulatory activity on in vitro cartilage cell homeostasis, focusing on chondro-progenitor cells (CPCs). Hence, in vivo setting was mimed by performing cartilage explant cultures derived from human biopsies in PL- or FBS-supplemented medium. In particular, this study sustains that articular cartilage from joints of patients with ongoing OA contains chondro-progenitor cell population that can be mobilized and subsequently ex vivo recruited from intact tissue by treatment with PL. Conversely, no cell population was recruited when cartilage chips were cultured in medium with FBS. This is the reason why a direct comparison between CPCs-FBS and CPCs-PL was missed in order to investigate the biology of this cell type.

Several studies have already confirmed the presence of stem/progenitor cells in articular cartilage and revealed that their outgrowth needs to be induced as occurs in pathologic condition or by application of external stimuli and trauma [56,57,58]. PL showed to be a powerful stimulus in this context since it is a cocktail of platelet released-growth factors, cytokines and chemokines—i.e., PDGF, IGF or SDF-1α (CXCL12)—widely studied for their multiple effects in the field of regenerative medicine, including chemoattraction of human progenitor cells [59,60,61].

In order to identify distinctive traits of CPCs among mature cartilage cell population, we compared cells isolated from cartilage chips cultured in presence of PL (CPCs-PL) with ACs enzymatically released from the surrounding tissue and expanded in PL. In parallel, we considered also the comparison between the latter and the AC counterparts maintained in culture with FBS.

Due to the high content in mitogenic growth factors, platelet products exert a strong proliferative stimulus on several in vitro cell cultures, as previously reported [14,15,30]. In this study, the proliferative trend demonstrated by doubling number and cyclin D1 protein levels were markedly higher in ACs-PL compared to ACs-FBS and similar between ACs-PL and CPCs-PL. These results are in accordance with previous works, where PL treatment on partially growth-arrested chondrocytes is reported as responsible for their cell cycle re-entry and strong mitogenic response [14,30]. Moreover, we further showed that the increase in proliferation is concomitant with a senescence attenuation in ACs-PL, at a similar level to that of CPCs-PL. In general, a subset of senescent cells is characterized by cell cycle irreversible arrest; meanwhile, quiescent and slowly proliferating cells can restart or increase proliferation after mitogenic stimulation [62,63]. Thus, our results reconfirm that PL is able to induce the re-entry of quiescent ACs into the cell cycle progression with consequent decrease of the senescent cell fraction no longer able to replicate. However, a recent hypothesis suggested senescence as a not-terminal cell event [64], thus platelet derivatives could even induce reversion from a senescent state. Indeed, it was shown that PRP is responsible for senescence recovery as well as induction of stem cell re-proliferation and differentiation in aged mice [65]. In general, our findings sustain that PL is suitable for expansion of chondrocytes intended for therapeutic applications, especially since senescent cartilage cells are involved in the development of OA in joints and their clearance may attenuate its progression promoting a pro-regenerative milieu [66].

Presence of MSC-related surface markers in CPCs supports their progenitor nature as described in previous reports [36,67]; meanwhile, their expression in ACs was due to phenotypic change known as dedifferentiation during monolayer culture [41]. CPCs-PL exhibited lower expression of CD106 than ACs-PL, which is in line with the results reported in a recent study [68] showing negativity for this chondrocyte marker in MSCs-like progenitors isolated from OA cartilage. Interestingly, CPCs-PL were consistently positive for CD166, whose co-expression with CD105 was reported in both bone-marrow-derived and cartilage mesenchymal progenitor cells [10,12]. Since the percentage of CD166^+^ cells increased in ACs-PL compared to ACs-FBS and results were similar to those found in CPCs, it suggests that PL is able to bring mature ACs to an earlier differentiation stage.

Aspects mostly highlighting the progenitor nature of CPCs-PL in this work were both their clonogenic potential and nestin expression. Interestingly, enzymatically digested ACs, unable to form colonies, acquired this ability when they were switched from FBS- to PL-supplemented medium. Articular cartilage contains a subpopulation of self-renewing cells, mainly concentrated in the superficial layer [9,57], but it was also reported that cells with stem cell phenotype can emerge even from fully differentiated chondrocytes [69]. Despite ACs used in this study being quiescent and mature, PL stimulation seemed to bring out certain features of stemness.

Nestin is a cytoskeletal protein of type VI intermediate filaments known to be expressed in proliferating and migrating stem/progenitor cell subsets in several human tissues [43], both during embryonic development [70] and after injuries in adulthood [26]. Our results are in line with the study of Fellows et al. [71], which detected nestin-labelled cells in human articular cartilage. An upregulation of this marker was detected in CPCs-PL compared to the weak expression observed in ACs-PL. However, self-renewing cells in colony units of ACs-PL were positive to nestin as well as in CPCs-PL. Nestin-expressing cells compose a quiescent reserve in adults that, if properly reactivated, is able to proliferate, differentiate and migrate. These events are triggered after injury, implying the involvement of these cells in processes of tissue regeneration [26], although the precise mechanisms have not been understood yet. Therefore, given the involvement of PL in such events, including the transient activation of the inflammatory cascade during tissue regeneration, some of its described effects could be due to activation and amplification of a subset of nestin-positive cells within the cartilage. Since CPCs-PL and ACs-PL were very similar in terms of proliferation, senescence levels and potential to form colonies, our findings suggest that PL retrieved a mixed cell population from cartilage tissue enriched in chondro-progenitors among committed cells.

Considering the features of CPCs-PL, which showed high proliferation capacity and stem/progenitor behaviour, we analysed their chondrogenic potential in view of a future therapeutic application. Generally, adult chondrocytes are characterized by a finite capability to form stable cartilage in vivo, that is gradually lost during in vitro monolayer culture [72]. ACs-PL displayed very low expression of type II collagen and a similar level of type I collagen in comparison to ACs-FBS, as previously reported [14]. CPCs-PL showed instead a slight but statistically significant increase in type II collagen expression compared to ACs-PL and a similar level of type I collagen. Such a trend in collagen distribution among cells considered chondro-progenitors is also confirmed by previous works [56,68]. Together with this change in collagen levels, a statistically significant difference was also found in the expression of the master regulator of chondrogenesis *SOX9* [73] among the three experimental groups, with a decrease in ACs-PL compared to ACs-FBS and a maintenance in CPCs-PL compared to ACs-PL. Indeed, it has been previously reported that chondro-progenitors showed decreased *SOX9* expression compared to fully differentiated chondrocytes, but it is maintained over extended monolayer culture passages together with chondrogenic potential [74]. Finally, Pereira et al. [14] have already demonstrated that PL, contrary to animal supplement, supports chondrocyte expansion preserving their chondrogenic phenotype both in vitro and in vivo.

In agreement with previous reports [36,56], CPCs-PL, when properly induced, underwent chondrogenic, osteogenic and adipogenic differentiation, capabilities not usual in articular cartilage. Such a trilineage differentiation potency is a typical property of adult MSCs, and thus it could be considered a distinctive trait of CPCs compared to normal chondrocytes, although adipogenesis is controversial and sometimes reported as limited or not inducible in these cells [57,68,75]. However, ACs-FBS and ACs-PL, derived from enzymatically digested tissue in toto, showed trilineage differentiation potency in our hands, probably because these two populations already contain the subset of progenitor/stem cells, as also reported by Alsalameh et al. [12].

In orthopaedics, several trials have been conducted to test the potential application of platelet derivatives, although their contributions still need to be clarified. Proteomic studies have shown a plethora of protein-based bioactive factors contained in platelet products [76,77]. Among them, released growth factors, such as TGF-β, PDGF, IGF, and bFGF (basic fibroblast growth factor), from platelets were shown to promote cartilage matrix synthesis and chondrogenic potential [78].

The strategies for progenitor cell recruitment/enrichment could be exploited for in situ reparative therapies based on platelet products, as previously demonstrated by Siclari et al. [20], or in a conventional tissue engineering approach since CPCs have shown multiple advantages compared to ACs and even to BM-MSCs for scaffold-assisted cartilage regeneration, as recently reported by [79]. In this perspective, we reported ectopic cartilage formation after implantation of expanded PL-recruited progenitors (CPCs-PL) in nude mice. Neo-formed tissue showed a typical metachromatic staining, and it was type II collagen positive and type X collagen negative and thus ascribable to a hyaline-like cartilage without hypertrophy signs. Therefore, migrating cells from articular cartilage under PL treatment (CPCs-PL) can be further considered a population consisting of chondro-progenitors since they were shown to be preferentially committed towards chondrogenic rather than osteogenic lineage in vivo.

Characterization of the conditioned media from ACs-FBS, ACs-PL and CPCs-PL could help in understanding the effects of PL on cartilage-derived cells and its involvement in tissue dynamics. Indeed, human chondrocytes, both normal and osteoarthritic, produce chemokines and express a variety of chemokine receptors, suggesting that their autocrine/paracrine pathway within cartilage may be involved in its homeostasis and matrix remodelling [55]. Secretome derived from digested ACs contains a plethora of molecules, including not only chondrocyte factors but also angiogenic proteins, pro-inflammatory cytokines, hypertrophic differentiation markers and chemokines with chemoattractant activity on immune system cells. Since cells used in the present study derived from articular cartilage biopsies with ongoing inflammatory processes, the described secretory profile could reflect the long-lasting exposure to a pathological joint environment.

PL treatment induced increased release or de novo secretion of some pro-inflammatory cytokines and chemokines by ACs. Indeed, it is well known from previous works that PL exerts an initial strong pro-inflammatory activity resulting in NF-κB activation and secretion of pro-inflammatory cytokines, events that are transient in order to prime certain defensive and reparative tissue mechanisms, then it inhibits and promotes resolution of the inflammatory phase [14,15].

Interestingly, the secretory profile of CPCs-PL differed from that of ACs-PL for soluble components involved in bone and cartilage biology or correlated with hypertrophic phenotype. Among them, the Wnt pathway antagonist Dkk-1 was highly upregulated in CPCs-PL. Dkk-1 is highly expressed in cartilage, and it is required for chondrogenic differentiation of MSCs, redifferentiation and prevention of hypertrophy of chondrocytes [80]. Furthermore, a low amount of ST2 was also detected in CPCs-PL compared to ACs-PL. ST2 has recently been described as a chondrocyte differentiation regulator that promotes expression of hypertrophic markers after RUNX2 induction [50]. Finally, osteopontin (OPN), a protein that contributes to hypertrophic phenotype in chondrocytes [81], was lower in the CM of CPCs-PL compared to the CM of ACs-PL. Another advantage of CPCs-PL over ACs-PL is that the former cell type secreted higher amounts of CST3, an inhibitor of cathepsins. In OA cartilage with severe lesions, a reduced inhibitory activity of CST3 has been reported, suggesting that downregulation of CST3 contributes to articular cartilage damage [82]. DPPIV release was also significantly higher in CPCs-PL than in ACs-PL. In a murine model of collagen-induced arthritis (CIA), it has been demonstrated that DPPIV injection decreases the overall extent of inflammation and articular damage around the arthritic joint and periarticular tissue [83].

Moreover, PL markedly increased the release of CCL5 (RANTES) by both ACs-PL and CPCs-PL, suggesting that PL exposure may promote the migration of joint resident cells towards damaged areas where they could engraft and take part in regenerative processes, as previously observed in degenerated intervertebral disk, where annulus fibrosus cells were mobilized in response to this secreted chemokine [84]. CCL5 has been reported to be involved also in MSCs recruitment [54], and thus it is considered attractive for regenerative therapies. As already reported [85], high levels of CXCL12/SDF-1α in chondro-progenitor secretome may indicate their ability to mobilize other endogenous progenitor cells within injured articular joints and recruit immune cells in order to help in mediating tissue repair.

A recent study has confirmed that chondro-progenitors are more suitable candidates for therapeutic treatment of meniscus injury than BM-MSCs since they resisted to hypertrophic differentiation during tissue repair [86]. Similarly, and according with our findings, we can suggest that PL-induced activation of such CPC population able to counteract cell hypertrophy within articular cartilage could be considered beneficial in therapeutic treatments for OA. However, further and more focused studies will be needed to investigate our hypothesis.

Although their role in cartilage disorder (OA) progression has not yet been clarified, chondro-progenitor cells has shown to be responsive to injury. In this study, we demonstrated that they are able not only to release into the surrounding environment chemoattractant factors but also to actively migrate in response to signals coming from inflamed chondrocytes. This is in accordance with the demonstrated capability of CPCs to migrate across the articular surface to sites that have suffered trauma and therefore display an ongoing inflammation [57]. These findings suggest that these mobilized cells may organize a kind of regenerative response in an attempt to restore the perturbed cartilage homeostasis in a damaged joint. Finally, PL exerted a general strong chemotactic action on the entire cartilage cell population, including mature chondrocytes (ACs) that usually are considered not provided with a mobile phenotype. Conversely to culture with FBS supplemented medium, ACs exposed to PL, probably as a result of changes in cell morphology and cytoskeletal rearrangements, acquired enhanced motility and were able to close a wound scratch in vitro, suggesting the possibility of using PL to promote their involvement in the overall tissue reparative process.

## 5. Conclusions

In conclusion, as reported previously, articular cartilage contains a reserve of progenitors, but since the tissue shows a limited regenerative potential, they seem to be not suitable to organize a proper resolving response after injury. Platelet products attract much interest for their intrinsic capacity to induce endogenous reparative and regenerative mechanisms when administered both in vitro and in vivo. This study demonstrates that PL exerts stimulant effects on articular cartilage, such as the promotion of chondrocyte proliferation, cell mobilization and activation of nestin-expressing progenitors. In particular, PL-recruited progenitor cells (CPCs-PL) are able to migrate in response to inflammatory stimuli, show paracrine activity in attracting other cells (ideally toward injured sites) and display high chondrogenic potential and resistance to hypertrophy. Thus, they might replace damaged chondrocytes in a compromised cartilage environment, thus representing a promising target for future therapeutic approach for cartilage disorders.

## Figures and Tables

**Figure 1 cells-09-01052-f001:**
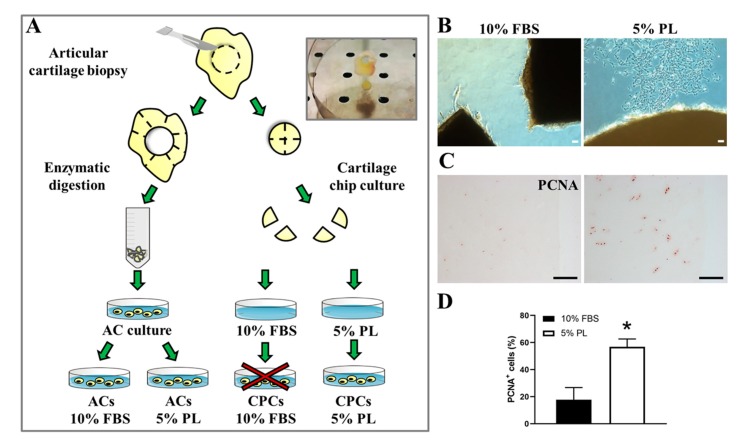
Experimental design of cell cultures (articular chondrocytes (ACs) and chondro-progenitors (CPCs) from human articular cartilage biopsies. (**A**) Representative illustration of biopsy handling to obtain ACs culture and cartilage chip culture. (**B**) Optical images of cartilage chips after 15–20 days in culture with cells coming out to the medium supplemented with platelet lysate (PL) versus fetal bovine serum (FBS) and (**C**) representative immunohistological distribution of proliferating cell nuclear antigen (PCNA)-positive cells inside tissue in both culture conditions (*N* = 3). (**D**) Histogram showing the percentage of PCNA-positive cells in cartilage chips maintained in culture with FBS or PL. Data are represented as mean ± SEM (*N* = 3, * *p* < 0.05 versus ACs 10% FBS by Student’s *t*-test analysis). All scale bars correspond to 100 μm.

**Figure 2 cells-09-01052-f002:**
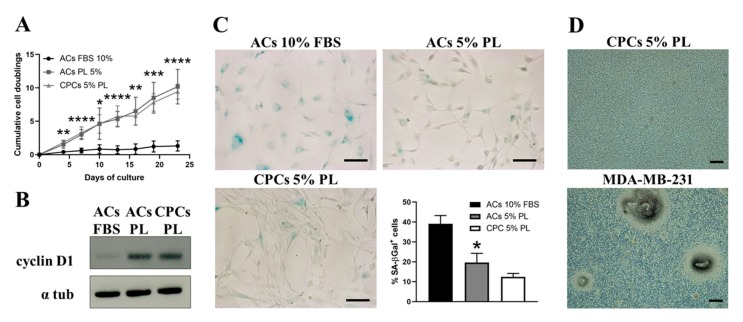
Growth rate and senescence profile of ACs and CPCs and in vitro tumorigenesis test on CPCs. (**A**) Growth kinetics plotted as number of cell duplications versus time of culture: black line indicates ACs expanded in presence of 10% FBS, grey line for ACs in 5% PL and light grey line for CPCs in 5% PL. Data are represented as mean ± SD (*N* = 6, * *p* < 0.05, ** *p* < 0.01, *** *p* < 0.001 and **** *p* < 0.0001 versus ACs 10% FBS at the same time point by Student’s *t*-test analysis). (**B**) Representative images of the detection of cyclin D1 and α-tubulin as control by Western blot in cell lysates from all the examined groups (*N* = 3). (**C**) Representative images of ACs 10% FBS, ACs 5% PL and CPCs 5% PL stained for senescence-associated SA-βgal and histogram showing the percentage of positive cells. Data are represented as mean ± SEM (*N* = 5, * *p* < 0.05 versus ACs 10% FBS by Student’s *t*-test analysis). (**D**) Anchorage-independent growth in methylcellulose for CPCs (upper panel) and human MDA-MB-231 breast cancer cells as positive control (bottom panel) (*N* = 3). All scale bars correspond to 100 μm.

**Figure 3 cells-09-01052-f003:**
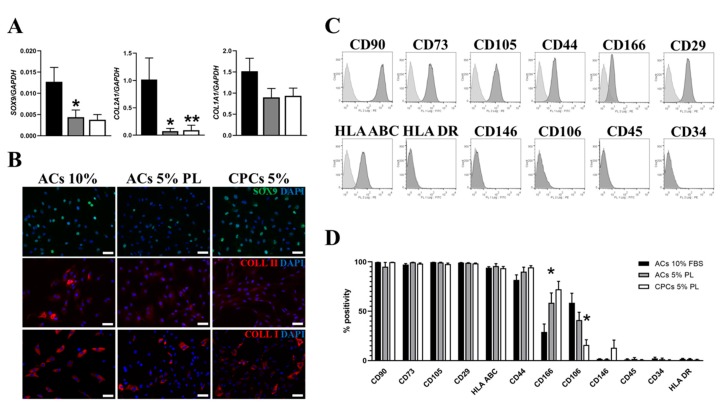
Analysis of chondrogenic and cell surface markers in ACs-FBS, ACs-PL and CPCs-PL. (**A**) Expression profile of the chondrogenic markers *SOX9*, *COL2A1* and *COL1A1* in ACs 10% FBS, ACs 5% PL and CPCs 5% PL determined by qRT-PCR. Data are represented as mean ± SEM (*N* = 7, * *p* < 0.05 versus ACs 10% FBS and ** *p* < 0.01 versus ACs 5% PL by Mann–Whitney test analysis). (**B**) Immunofluorescence staining for SOX9 (upper panel), type II collagen (middle panel) and type I collagen (bottom panel) in the three experimental groups. (**C**) Representative flow cytometry characterization of CPCs for a set of typical surface markers: light grey peaks indicate the isotype control staining and dark grey peaks indicate the antibody staining. (**D**) Histogram reporting the percentage of positive ACs 10% FBS, ACs 5% PL and CPCs 5% PL for each tested marker in flow cytometry experiments; data are represented as mean ± SEM (*N* = 3, * *p* < 0.05 versus ACs-FBS for CD166 and versus ACs-PL for CD106 by Student’s *t*-test analysis). All scale bars correspond to 100 μm.

**Figure 4 cells-09-01052-f004:**
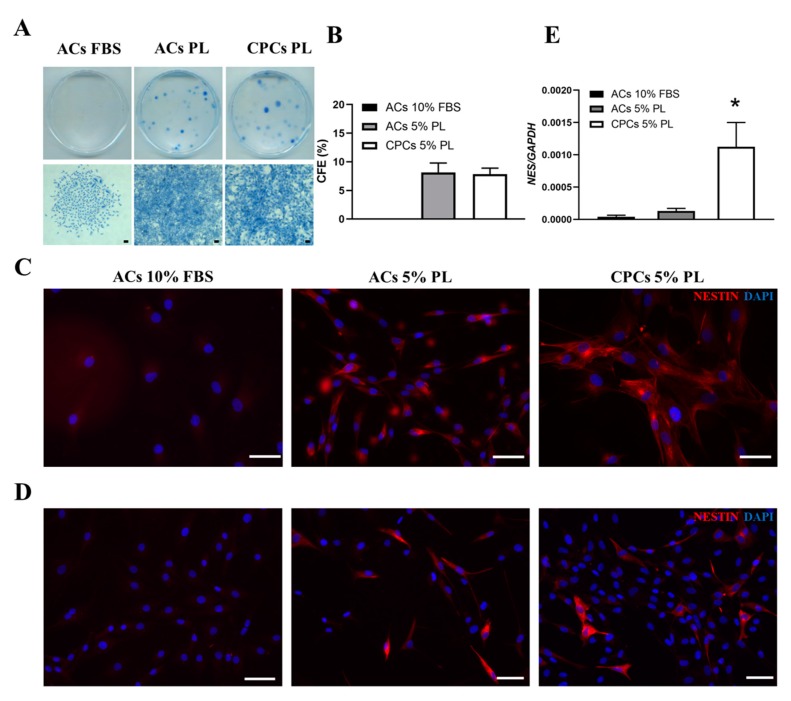
Clonogenic potential and detection of nestin expression in ACs-FBS, ACs-PL and CPCs-PL derived from articular cartilage tissue. (**A**) Colony-Forming Unit Fibroblast (CFU-F) assay for ACs isolated in 10% FBS, ACs treated with 5% PL and CPCs recruited by 5% PL from cartilage chips: cell colony staining in culture dish (upper panel) and magnification of stained colony cells (bottom panel). (**B)** Histogram reporting the colony forming efficiency (CFE) calculated for the three experimental groups. Data are represented as mean ± SEM (*N* = 10, no statistically significant difference by Student’s *t*-test analysis). (**C**) Immunofluorescence staining for nestin in CFU-F cells/dish of the three examined groups. (**D**) Immunofluorescence staining for nestin in the three experimental groups. (**E**) Expression level of the stem cell marker nestin (*NES*) in ACs 10% FBS, ACs 5% PL and CPCs 5% PL determined by qRT-PCR. Data are represented as mean ± SEM (*N* = 5, * *p* < 0.05 versus ACs 5% PL by Mann–Whitney test analysis). All scale bars correspond to 100 μm.

**Figure 5 cells-09-01052-f005:**
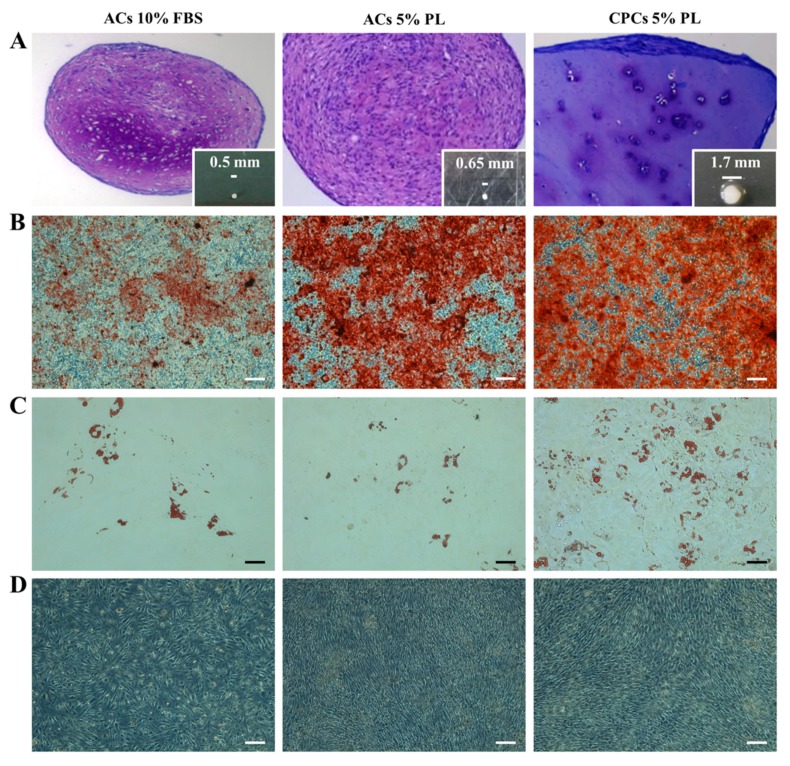
Multi-lineage differentiation potential of ACs-FBS, ACs-PL and CPCs in vitro. (**A**) Toluidine blue staining of three-dimensional pellets formed by ACs 10% FBS, ACs 5% PL and CPCs 5% PL after 21 days of chondrogenic differentiation in vitro; insets show pellet appearance and size. (**B**) Alizarin red staining for the three experimental groups after 21 days of osteogenic culture in vitro. (**C**) Oil red O staining for the three experimental groups after 21 days of adipogenic culture in vitro. (**D**) Uninduced cell controls. All scale bars correspond to 100 µm (*N* = 3).

**Figure 6 cells-09-01052-f006:**
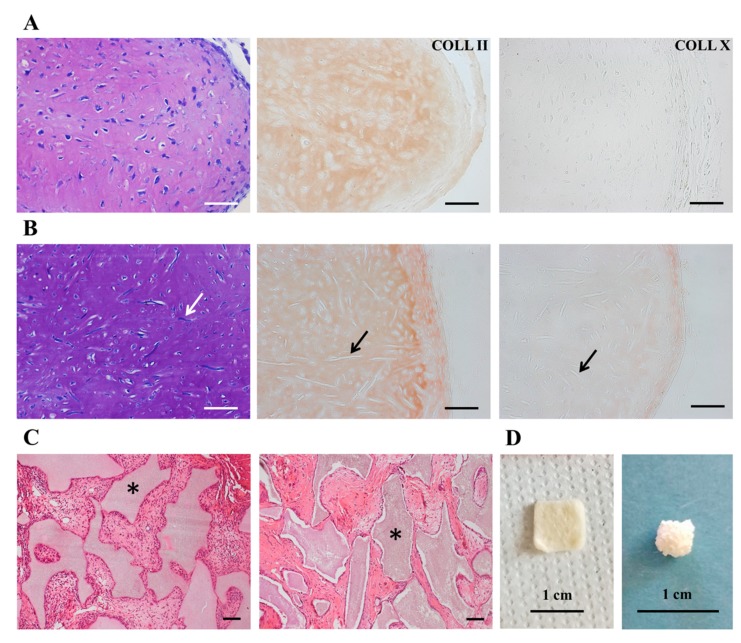
In vivo ectopic chondrogenesis and osteogenesis assay of CPCs. (**A**,**B**) Histological analysis of ectopic cartilage formed in vivo after subcutaneous implantation of CPC-pellets (**A**) and CPCs-seeded biomaterials (**B**) in nude mice; from left to right: toluidine blue staining, type II and type X collagen stainings. Arrows point out the remnants of polyglycolic-acid (PGA)-scaffold fibers. (**C**) Histological analysis by hematoxylin/eosin staining of ectopic tissue formed in vivo after subcutaneous implantation of ACs-FBS (left) or CPCs (right)-seeded osteoinductive scaffolds in nude mice. Asterisks point out the biomaterial. (**D**) Representative view of selected scaffolds, polyglycolic-acid–hyaluronan (PGA-HA) mat (left) and hydroxyapatite/β-tricalcium phosphate (HA/β-TCP) assembled granules (right). Scale bars correspond to 50 µm (upper panel), 100 µm (middle and bottom panels) and 1 cm.

**Figure 7 cells-09-01052-f007:**
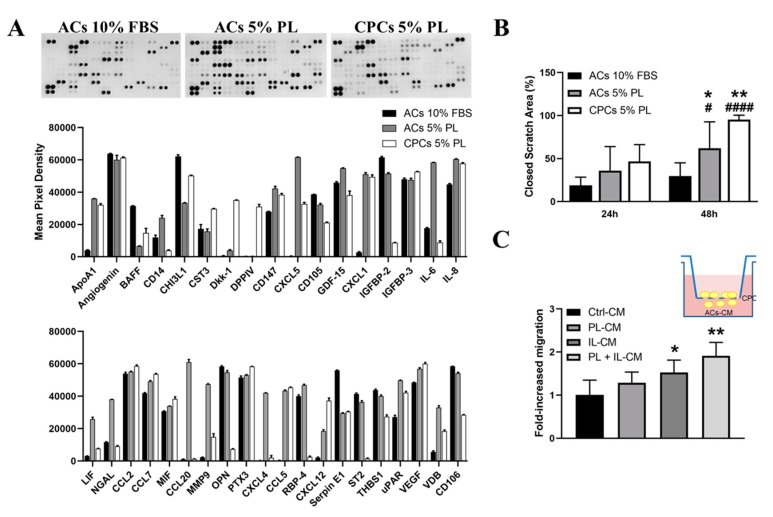
In vitro analysis of ACs and CPCs secretory profiles, migratory capability and CPC chemotaxis in an osteoarthritis (OA) mimetic system. (**A**) Quantification of the mean pixel density for each identified cytokine in the three experimental groups. Error bars represent the uncertainty in measurement; pictures on the top shows the cytokine array membranes of CM of ACs—10% FBS, ACs—5% PL and CPCs—5% PL, from left to right. (**B**) “Closed scratch area (%)” calculated by TScratch software at each time points in all three experimental groups. Data are represented as mean ± SD (*N* = 3, * *p* < 0.05 and ** *p* < 0.01 versus ACs-FBS or ACs-PL at the same time point; # *p* < 0.05 and #### *p* < 0.0001 versus 24 h in the same experimental group by Kruskal-Wallis with Dunn’s multiple comparison test). (**C**) Quantification of CPC migration in the presence of ACs-CM obtained in four different conditions: control (CTRL-CM, 10% FBS culture), PL-treatment (PL-CM, 5% PL culture), inflammatory stimulus (IL1-CM, IL1-β culture) and PL-treatment plus inflammatory stimulus (PL+IL1-CM, 5% PL plus IL1-β culture). Data (mean ± SD) are reported as fold-increased migration as compared to control, set to 1 value (*N* = 3, * *p* < 0.05 and ** *p* < 0.01 versus CTRL-CM by one-way ANOVA analysis).

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
