# Peer review of "Progenitor Cells Activated by Platelet Lysate in Human Articular Cartilage as a Tool for Future Cartilage Engineering and Reparative Strategies"

_cells, 2020, doi:10.3390/cells9041052_

Round 1

Reviewer 1 Report

In their manuscript, Carluccio et al., show that Platelet lysate activate chondro progenitor cells in terminally differentiated cartilage tissue. They show that Platelet lysate enhance lineage differentiation of chondro-progenitors with high proliferation and nestin stem cell marker expression.

The study is well performed and article in general is well-written.

I have following concerns.

  1. Figure 1, the authors should provide quantification for PCNA staining
  2. In Figure 2, could be good to perform these experiments in dose-dependent manner including several increasing concentration of platelet lysate.

The authors only analyze the senescence; what is rate of apoptosis? In many cancer cell types, B-gal stains also dormant cells. Could be also b-gal stained cells not only senescent cells?

  1. To what the ratio interest gene/GAPDH corresponds, as indicated in all quantitative PCR experiments? How it is calculated, delta Ct ?.. They should normalize to the control group. Ratio makes non-sense
  2. Figure 4, quantification for the CFU-F assay is required

The authors also should discuss how platelet lysate may enhance these effects. They must check publications described secretome of platelets and propose hypothesis which molecules factors stored in platelet granules may stimulate differentiation.

Author Response

Reviewer #1

In their manuscript, Carluccio et al., show that Platelet lysate activate chondro progenitor cells in terminally differentiated cartilage tissue. They show that Platelet lysate enhance lineage differentiation of chondro-progenitors with high proliferation and nestin stem cell marker expression.

The study is well performed and article in general is well-written.

We thank the Reviewer for her/his appreciation of our work. We addressed in depth all her/his comments to further improve quality.

I have the following concerns.

  1. Figure 1, the authors should provide quantification for PCNA staining

Following the suggestion of the Reviewer, we added a histogram (Figure 1D, page number 9) in the revised manuscript reporting the PCNA positive cell percentage (%) calculated on cartilage chip sections who underwent immunohistochemistry staining for this proliferation marker. We found by this quantitative analysis a statistically significant increase of PCNA stained cells in tissue fragments treated with PL compared to FBS, confirming the proliferative stimulus induced by the exposure to platelet derivatives. We integrated section 2.12 in Material and Methods (page number 7, lines 294–296) with technical issues. Furthermore, we reported obtained percentages for both tested groups in section 3.1 of Results (page number 8, line 362).

  1. In Figure 2, could be good to perform these experiments in dose-dependent manner including several increasing concentration of platelet lysate.

We thank the Reviewer for the comment which offers us the opportunity to better explain the experimental conditions chosen in the work. In general, the final concentration of PL in cell culture media ranges from 1% to 60%, with 5% to 10% being the most common concentration used. This percentage will depend on the starting platelet concentration, and differs among studies as well as depends on tested cells (see Burnouf, Biomaterials, 2016). In our preliminary experiments we have already tested several PL concentrations (from 2.5% to 10%) on chondrocytes and other cell cultures (including mesenchymal stem cells, skin fibroblasts and osteoblasts). What we have always observed was that 5% PL is the maximum effective concentration in terms of growth stimulation and higher doses have similar or even adverse effects (see Zaky, J Tissue Eng Regen Med., 2008; Muraglia, Platelets, 2014 [28]). We also performed preliminary tests on cartilage chip cultures with a range of increasing PL concentration (from 2.5% to 10%) and even with non-lysed PRP. Once again we recognized 5% PL as the recommended dose to obtain CPC recovery and its reproducibility. Consequently, we think that other concentrations, lower or higher than 5%, can’t allow to value the real contribution of PL in tissue or cell dynamics intended for regeneration. Furthermore, PL added to culture medium at 5% concentration approximately corresponds to the highest physiological platelet dose in human blood, without heparin addition. Moreover, high concentration of PL gives rise to culture medium gelation, issue resolvable only with supplementation of factors (like heparin) that may change cell responses (Muraglia, Front Bioeng Biotechnol., 2017 [29]). Data heterogeneity on dose-related effects of platelet products is common in the scientific literature due to the lack of standardization procedures for its preparation and consequently variability in growth factor concentrations. We provided this technical information in 2.1 section of Materials and methods (page number 3, lines 109–112).

  1. The authors only analyze the senescence; what is the rate of apoptosis? In many cancer cell types, B-gal stains also dormant cells. Could be also b-gal stained cells not only senescent cells?

We thank the Reviewer for her/his comment. We detected the activity of β-galactosidase (β-gal) in ACs and CPCs at pH 6 when it is considered a biomarker of replicative senescent cells (see Dimri, Proc Natl Acad Sci U S A., 1995). The cell cycle arrest occurred in these cells frequently correlates to aging, thus they are destined to exhaust their replicative potential in tissues during the lifespan. Osteoarthritis (OA) is a cartilage disease with aging as one of the major factor risks and senescent β-gal positive cells were identified close to the osteoarthritic lesions of pathological cartilage as opposed to healthy tissue (see Price, Aging Cell., 2002). Furthermore, recently a divergent sub-population of cartilage-progenitor cells with reduced proliferative potential and senescence signs were isolated from OA cartilage (see Fellows, Sci Rep., 2017 [34]), partially justifying the inability of tissue in self-repairing. Since our mission is to demonstrate the validity of platelet product use as therapeutic agent for cartilage diseases, we wondered if a difference in senescent cell portion could be detected in ACs-PL and CPCs-PL in comparison with normal ACs from elderly donors, and especially in conjunction with the PL-induced mitogenic stimulus. The reduction in β-gal positive cell percentage that we found can be explained as follows. Senescence was traditionally considered as irreversible cell growth arrest and in this perspective, the percentage of senescent cells decreases due to the proliferation of the remaining cells (not senescent) under PL stimulation. However, a recent hypothesis suggested senescence as a not terminal cell event (see Lee, Nat Cell Biol., 2019 [64]), at least for cancer cells, and in this perspective platelet derivatives may also induce reversion from the senescent state. Indeed, it was shown that Platelet Rich Plasma (PRP) is responsible for senescence recovery as well induction of stem cell re-proliferation and differentiation in aged mice (see Liu, Biomaterials, 2014 [65]). In general, stem/progenitor cells are thought to be retained quiescent over prolonged periods of dormancy in the body (see Dexheimer, PLoS One, 2011), but can be stimulated to enter the cell cycle, proliferate and undergo differentiation in a repair situation by external cues, as treatment with platelet derivatives may probably act.

Lastly, in contrast to apoptosis, senescent cells are stably viable and secrete soluble factors, which are known as the senescence-associated secretory phenotype (SASP), in the surrounding environment. (see Childs, EMBO Rep., 2014). Senescence and apoptosis are considered alternative cell fates and it is difficult to understand what decides the execution of one or the other program. In this regard, senescent cells are resistant to apoptosis (see Sasaki, Mech Ageing Dev., 2001; Yosef, Nat Commun.,2016).

Among the beneficial effects of platelet derivatives, chondroprotection could probably also occur by reducing the apoptosis rate and a recent study seems to confirm this tendency (see Moussa, Exp Cell Res., 2017 [38]). In our hands, we have already started to preliminarily study apoptosis in ACs and CPCs by the detection of cleaved caspase-3, which represents a reliable marker for cells that are dying by apoptosis (see Porter, Cell Death Differ., 1999) in immunofluorescence experiments. Contrary to Moussa et al. we did not detect any evident difference in cleaved caspase-3 between the three examined cell groups (ACs-FBS, ACs-PL, and CPCs-PL), rather its levels appeared quite low. We added representative images of this very preliminary analysis in Figure S1 of Supplementary Materials and technical detail in section 2.8 of Materials and Methods (page number 5, lines 213–214) of the revised manuscript. Furthermore, we think that we probably need to induce in our cells stress or apoptotic signals in order to trigger and subsequently detect an anti-apoptotic effect of PL. Unfortunately, given the current situation of exceptional emergency due to the Covid-19 outbreak, we are not able to access any laboratory to perform additional experiments. We recalled some of these considerations in section 3.2 of Results (page numbers 9–10, lines 390–397) and in Discussions (page number 17, lines 649–652).

  1. To what the ratio interest gene/GAPDH corresponds, as indicated in all quantitative PCR experiments? How it is calculated, delta Ct ? They should normalize to the control group. Ratio makes non-sense.

Following the Reviewer’s comment, we clarified that ratio in our data analysis corresponds to the expression of the gene of interest (e.g. SOX9, Type II Collagen) divided by the expression of GAPDH in the same sample. More in detail, it corresponds to 2-ΔCt Gene of interest (e.g. SOX9) of sample 1/2-ΔCt GAPDH of sample 1; 2-ΔCt Gene of interest (e.g. SOX9) of sample 2/2-ΔCt GAPDH of sample 2; 2-ΔCt Gene of interest (e.g. SOX9) of sample 3/2-ΔCt GAPDH of sample 3; and so on, being GAPDH the housekeeping gene selected for normalization. Delta Ct was calculated as Ct of sample 1 - Ct of sample 1; Ct of sample 2 - Ct of sample 1; Ct of sample 3 - Ct of sample 1; Ct of sample 4 - Ct of sample 1; and so on. Therefore, we took Ct of sample 1 as a "reference" for all the samples. We think that the normalization to the control group does not give further information since we are comparing only two groups: ACs 10% FBS (Control) vs. ACs 5% PL or ACs 5% PL (Control) vs. CPCs 5% PL. We also believe that the ratio “Gene of interest/GAPDH” is absolutely needed because gene expression must be always normalized by a housekeeping gene. This method of normalization is widely accepted as an appropriate normalization method to avoid the influence of differences in the extraction of mRNA between samples and performance of reverse transcription and PCR itself (see Huggett, Genes Immun., 2005; Kozera and Rapacz, J Appl Genetic,s 2013). We recalled these specifications in 2.7 section of Materials and Methods (page number 5, lines 201–203) and clarified the two types of comparison that we chose as appropriate in studying the three cartilage cell populations in Discussion (page number 17, lines 624–626 and 633–636).

Figure 4, quantification for the CFU-F assay is required

Following the suggestion of the Reviewer, we added a histogram (Figure 4B, page number 12) in the revised manuscript reporting the Colony Forming Efficiency (CFE %) for each of the examined experimental groups. CFE % is calculated as a percentage of the ratio between cell colony number and plated cell number. We integrated the 2.9 section in Materials and Methods (page number 5–6, lines 236–237) with indications on the analysis performed. The comparison showed a similar clonogenic potential between CFE% of ACs-PL and CPCs-PL, that is absent in ACs-FBS, and suggested a role of PL treatment in the emergence of stem cell-like features among cartilage-derived cells. We reported these results in section 3.4 in Results (page number 11, lines 464–466), specifying also the number of the performed counts for each examined cell groups.

The authors also should discuss how platelet lysate may enhance these effects. They must check publications described secretome of platelets and propose hypothesis which molecules factors stored in platelet granules may stimulate differentiation.

We thank the Reviewer for her/his suggestion. Effects of platelet lysate (PL) on treated cells is a certain consequence of the action exerted by the wide range of molecules released from activated platelets. Furthermore, the interest in platelet derivatives in therapeutic strategies lies in their potential to re-activate endogenous mechanisms of tissue regeneration, including cell proliferation and stem cell differentiation. Proteomic studies have shown a plethora of molecules, more than 1.500 (see Boswell, Arthrosc. - J. Arthrosc. Relat. Surg., 2012 [76]) and up to 5000 (see Freson, Academic Press., 2019 [77]) protein-based bioactive factors. Among them, released growth factors such as transforming growth factor-β (TGF-β), PDGF (platelet-derived growth factor), IGF (insulin-like growth factor), and bFGF (basic fibroblast growth factor) from platelets showed to promote cartilage matrix synthesis and chondrogenic potential (see Kabiri, Adv Biomed Res., 2014 [78]). The content in TGF-β could be particularly beneficial in the cartilage field since it plays a significant role in regulating chondrocyte homeostasis from early to terminal stages (see Wang, Birth Defects Res. Part C - Embryo Today Rev., 2014). In particular, TGF-β acts as an inhibitor of terminal hypertrophic differentiation in post-natal chondrocytes and promotes their anabolism, by enhancing matrix production, cell proliferation, and osteochondrogenic differentiation. Furthermore, in vivo short-term intra-articular injections showed beneficial effects on osteochondrogenesis (see Grimaud, Cytokine Growth Factor Rev., 2002). Another molecule reported as abundant in platelet secretome and potentially involved in cartilage homeostasis is thrombospondin-1 (THBS1) (see Della Corte, Platelets, 2008). This protein has anti-angiogenic and anti-fibrogenic effects (see Dobaczewski, J. Mol. Cell. Cardiol., 2010) and recently its role of chondroprotection has been described in OA animal model (see Maumus, Front. Immunol., 2017). We think that all these aspects are adequate to justify the performance of the progenitor cell population (CPCs) recovered from PL-treated cartilage in our work, but other mechanisms can be trigger by platelet products in tissue and cell dynamics due to their wide cargo. We recalled these considerations in the Discussion of the revised manuscript (page number 18, lines 712–715).

Reviewer 2 Report

Review manuscript: "MSC-like Progenitor Cells Activated by Platelet Lysate in Human Articular
Cartilage as a Tool for Future Cartilage Engineering and Reparative Strategies.”; Carluccio &
Martinelli et al.
Summary
Carluccio & Martinelli et al. studied the regenerative capacity of platelet lysate (PL) on primary
chondrocyte cultures and a chondroprogenitor-enriched population from ex vivo cartilage
cultures. They here report that PL activates a distinct stem cell niche and, therefore, can be
used as potential therapeutic strategy towards e.g. osteoarthritis (OA).
OA is an important disease which affects many people and a complete therapy still remains
elusive. Therefore, further basic and translational research studies focusing on underlying
mechanism of pathogenesis and potential therapeutic strategies are needed. Thus, the
manuscript would contribute towards solving an unmet clinical need.
In general, the manuscript is very well written. The introduction gives a good overview on the
topic for a wider readership; the methods are clearly described as well as the results which are
finally comprehensively discussed in the context of the current literature.
However, while reading the manuscript smaller technical weaknesses and questions for the
understanding arose. Therefore, I recommend a Minor Revision addressing the following
considerations:
General aspects
• Within the title, the authors claim that PL activates MSC-like progenitors while in the
abstract and in most of the manuscript, the authors mention chondroprogenitors
(please write it consistently). It would be much easier for the reader when the difference
or similarity is clearly stated, and one term is kept. Furthermore, there are clear
guidelines published on the characterizations of MSCs – it would be good to discuss
the similarities of the found cell population with those guidelines. Based on the
possibility to differentiate into all three lineages and presence or absence of different
surface markers the assumption “MSC-like” seems to be likely but should be perhaps
pointed out more clearly.
Technical aspects
• Unfortunately, the statistical analysis seems to be not appropriate and must be
adapted.
o It is not reasonable why the standard error mean (SEM) was used for all graphs
– the SEM can be used e.g. for qPCR data which are based on the mean of
duplicates or triplicates which is e.g. also the case for ELISAs. If all assays have
been performed as duplicates or triplicates and the mean was taken for
analysis, this should be stated precisely. Otherwise the standard deviation
should be indicated.
o Due to the low n-number (n=3 – most often), it is not possible to assume
Gaussian distribution (no normality test that works in this case). Based on the
lack of distributional certainty, non-parametric methods (Mann Whitney test if
unpaired; Wilcoxon sign ranked if paired; and Kruskal Wallis with Dunn’s
multiple comparison test for group differences) need to be applied.
o Since also non-parametric methods are not in themselves a solution to the issue
of small sample size, the limitations of interpretation should be clearer in the
discussion.
o In addition, it should be shortly explained how sample size calculation was
performed or why the sample size was kept so low. In the methods part it is
mentioned that n=20 samples were collected – which read-out parameter was
determined for which sample? Please also discuss the issue of heterogeneity
and the comparability between the samples/results. Where there any quality
measures applied? Or what could be potentially be used as quality measure?
Please elaborate on this point.
# Abstract
• The abstract is compact and precise.
• Line 23: Please clarify or change the word joint-progenitor cell population – joint-related
or resident?
• Line 24: Replace thanks with due
# Introduction
• The introduction is nicely written and gives a good overview on the topic for a wider
readership.
• Line 66/67 concerning the musculoskeletal system
# Materials and Methods
• It would be helpful for experts when the source of the cartilage (hip) would be directly
stated in the abstract or introduction.
• Why was no control group conducted for the CPCs (+ 10% FCS)?
• How much oxygen was provided? Have the authors thought about using hypoxic
conditions?
• For the sake of completeness, please provide the animal experiment application
numbers for the NOD/SCID mice and the CD-1 nu/nu mice experiments and state that
all experiments were approved by local authorities. Age and sex would be also good.
• Please shortly elaborate why GAPDH was used as housekeeper gene?
• Genes names/abbreviations should be written in italic – this applies also for the Result
section
# Results
• The results are clearly written, and the figures are overall understandable. There are
minor suggestions that could support the description of the findings.
• Fig. 1 legend – instead of Setting up you could take Experimental design of cell cultures
• Fig. 1A – This Figure is helpful and should be enlarged by including the next level of
experiments - ACs 5%PL and ACs 10% FCS and CPCs 5% PL
• Fig. 3 B – Please include the stained targets Sox9 Col II and Col I in or beside the
images in the given color (also Fig. 4B, D; Fig. 6 Col II and Col X). Since Sox9 is a
transcription factor and mostly expressed in the cell nucleus – co-illustration with
DAPI would be helpful.
• In addition, for the fluorescence images – overviews are quite helpful to support the
assumptions (if available).
• Fig. 7 A – Please split the graph in subgroups – otherwise the bars are not readable
or comparable; what do the images above stand for? Where are the error bars?
• Fig. 7 C – Pease adapt graph to the other graphs – what does fold-increase migration
mean? How was normalization performed?
# Discussion
• the discussion is really detailed and includes the main findings
• Line 708 what does CM mean?
• Can the authors relate any of the observations to the digestion which was performed
for AC isolation but not for CPC?

All the best and stay healthy!

Author Response

Reviewer #2

We thank the Reviewer for appreciating the experimental work that we presented in the manuscript. We addressed in depth all her/his comments as detailed below, in order to further increase quality.

However, while reading the manuscript smaller technical weaknesses and questions for the understanding arose. Therefore, I recommend a Minor Revision addressing the following considerations:

General aspects

  • Within the title, the authors claim that PL activates MSC-like progenitors while in the abstract and in most of the manuscript, the authors mention chondroprogenitors (please write it consistently). It would be much easier for the reader when the difference or similarity is clearly stated, and one term is kept. Furthermore, there are clear guidelines published on the characterizations of MSCs – it would be good to discuss the similarities of the found cell population with those guidelines. Based on the possibility to differentiate into all three lineages and presence or absence of different surface markers the assumption “MSC-like” seems to be likely but should be perhaps ,pointed out more clearly.

Following the suggestion of the Reviewer, we agreed to adapt the title of the revised manuscript and use the term “chondro-progenitors” referring to cartilage-derived cells activated by PL in the main text (Discussion, page number 17, lines 622). The aim of this decision was to not create doubt for the readers and stay consistent with the cell commitment showed in in vivo experiments and so with the message of our work. However, we are fully convinced about the “MSC-like” nature of the cell population that we characterized, since it showed some shared features with MSCs from the developmental origin, to ability in forming CFU-F (see Bianco, Cell Stem Cell., 2008), tripotent differentiation capacity and a certain superficial phenotype (see Dominici, Cytotherapy, 2006) and lastly the expression of Nestin marker (see Bernal and Arranz, Cell Mol Life Sci., 2018).  

Technical aspects

  • Unfortunately, the statistical analysis seems to be not appropriate and must be adapted.

o It is not reasonable why the standard error mean (SEM) was used for all graphs– the SEM can be used e.g. for qPCR data which are based on the mean of duplicates or triplicates which is e.g. also the case for ELISAs. If all assays have been performed as duplicates or triplicates and the mean was taken for analysis, this should be stated precisely. Otherwise the standard deviation should be indicated.

We thank the Reviewer for her/his comment. We clarified in section 2.17 of Materials and Methods (page number 8, lines 338–346) in the revised manuscript that SEM was adopted with means taken from duplicates/ triplicates and SD in the other case. The indication was recalled in the respective figure legends. When the values were mentioned in the main text (doubling number on page number 9, line 377; CSA% at page numbers 15–16, lines 584–588) the corrections for the measurement errors were introduced in the revised manuscript (SD instead of SEM).

o Due to the low n-number (n=3 – most often), it is not possible to assume Gaussian distribution (no normality test that works in this case). Based on the lack of distributional certainty, non-parametric methods (Mann Whitney test if unpaired; Wilcoxon sign ranked if paired; and Kruskal Wallis with Dunn’s multiple comparison test for group differences) need to be applied.

o Since also non-parametric methods are not in themselves a solution to the issue of small sample size, the limitations of interpretation should be clearer in the discussion.

Following the Reviewer comments, we performed normality tests on our data followed by non-parametric tests in case of the missed normal distribution for the statistical analysis as specified in section 2.17 of Materials and Methods (page number 8, lines 338–346). This intervention led to reasonable changes in some result arrangements as described below. The statistical significance comparison between proliferation curves (Figure 2A, page number 10) changed, but without altering the conclusions already deduced. The statistical significance in COL2A1 and SOX9 expressions for the chondrogenic phenotype analysis (Figure 3A, page number 11) among the examined groups changed. In this regard, a downregulation of SOX9 levels (already predictable in the previous version) emerged although it seems to not compromise chondrogenic potential of the examined cells. Finally, the statistical significance in time points and culture conditions changed in the scratch assay (Figure 7B, page number 16), where the difference in the motility of the examined groups were postponed to 48 hours, with ACs-PL taking the lead on ACs-FBS, and CPCs on ACs-PL as noted previously. The respective changes were reported in the graphs and figure legends, as well as in section 3.3 and 3.8 of the Results (page numbers 10 and 15; lines 422–428 and 584–588) and in Discussions (page number 18, lines 692–698).

o In addition, it should be shortly explained how sample size calculation was performed or why the sample size was kept so low. In the methods part it is mentioned that n=20 samples were collected – which read-out parameter was determined for which sample? Please also discuss the issue of heterogeneity and the comparability between the samples/results. Where there any quality measures applied? Or what could be potentially be used as quality measure? Please elaborate on this point.

We thank the Reviewer for the comment which offers us the opportunity to better explain the reasons for the experimental conditions adopted in the work. The gap between the number of collected biopsies and the sample size for each study is mainly due to the quantity of cartilage tissue not defective and homogeneous in appearance that it was possible to harvest from the surgical biopsies and subsequently split for enzymatic digestions and simultaneous organ cultures from each donor. Most of the experiments required a high number of cells to be performed and at the same time we consider worthwhile to manipulate cells at early culture passages (not more than P3) in order to not induce irreversible chondrocyte dedifferentiation (see Dell’Accio, Arthritis Rheum., 2001; Schulze-Tanzil, Osteoarthritis Cartilage, 2004). The issue of heterogeneity and comparability to which the Reviewer referred could be in our hands correlated to the degree of degenerative conditions of the patient cartilage tissue. The scientific literature reports studies in which progenitor frequency and proliferation increases in pathological cartilage (see Fellows, Sci Rep., 2017 [34]; Mankin, J Bone Joint Surg Am., 1971), while they are not influenced by donor age (see Fellows, Sci Rep., 2017 [34]; Williams, PLoS One, 2010[36]). We integrated section 3.1 in Results (page number 8, lines 363–364) with the just discussed observations.

# Abstract

  • The abstract is compact and precise.
  • Line 23: Please clarify or change the word joint-progenitor cell population – joint-related or resident?

Revised as requested (line 23).

  • Line 24: Replace thanks with due

Revised as requested (line 24).

# Introduction

  • The introduction is nicely written and gives a good overview on the topic for a wider readership.
  • Line 66/67 concerning the musculoskeletal system

Revised as requested (lines 66-67).

# Materials and Methods

  • It would be helpful for experts when the source of the cartilage (hip) would be directly stated in the abstract or introduction.

Revised as requested (lines 28, 89, 95).

  • Why was no control group conducted for the CPCs (+ 10% FCS)?

We thank the Reviewer for her/his comment. As we reported in the main manuscript (Discussion, page number 17, lines 624–626), the control group CPCs-FBS was not used for comparison with CPCs-PL due to the failed harvesting of cells in cartilage chip culture supplemented with 10% FBS. Although in sporadic cases, we noted some cell outgrowth from tissue fragments, they did not proliferate and could not be expanded in enough quantity and reasonable time to be tested in parallel with the other cell cultures. Consequently, ACs-PL has claimed the control group of CPCs-PL based on the uniformity of the growth conditions (exposure to PL), while ACs-FBS were considered in the comparison with ACs-PL based on the uniformity of isolation method (tissue digestion). We recalled this aspect in the figure 1A (page number 9) of the revised manuscript following the below Reviewer comment and in Discussion (page number 17, lines 624–626 and 633–636).

  • How much oxygen was provided? Have the authors thought about using hypoxic conditions?

We thank the Reviewer for her/his comment. Cell cultures in this work were performed in normoxic conditions, with atmospheric pO2 set around 20-21 % as reported in section 2.2 in Materials and Methods (page number 3, lines 136–137) in the revised manuscript. We are well aware of the importance of a hypoxic environment in cartilage homeostasis and biology and in a future perspective we hope to address our studies on the topic. Due to the lack of vasculature, oxygen tension within articular cartilage is estimated to range from approximately from 1% to 7% across the tissue (see Fermor, Eur Cell Mater., 2007). We couldn’t perform chondrocyte and cartilage cultures in this condition and simultaneously with the normoxic one because of the aforementioned difficulties in the sample management. Furthermore, given the current situation of emergency due to the Covid-19 outbreak, we are not able to access in our laboratory for setting up new experiments, as well as to obtain new biopsies from hospital orthopedics department. However, we can point out some starting points that may be useful in our future researches. In the aforementioned range of low oxygen tension, the action of HIF transcription factors plays a crucial role in cartilage physiology (see Araldi, Bone, 2010) and consequently also on the maintenance and gaining of chondrogenic phenotype in culture (see Meretoja, Biomaterials, 2013). In a recent work of our laboratory, it was highlighted an interesting induction of HIF-1α in articular cartilage chondrocytes by PL treatment in in vitro normoxic cultures (see Nguyen, J Tissue Eng Regen Med, 2018 [30]).

  • For the sake of completeness, please provide the animal experiment application numbers for the NOD/SCID mice and the CD-1 nu/nu mice experiments and state that all experiments were approved by local authorities. Age and sex would be also good.

Revised as requested (page numbers 4 and 6; lines 180–182 and 260, 272, 275–276).

  • Please shortly elaborate why GAPDH was used as housekeeper gene?

We thank the Reviewer for her/his comment. GAPDH gene is considered a stably expressed reference gene in the scientific literature and common utilized in cartilage and bone qPCR analysis (see He, Sci Rep., 2018 [33]). We recalled this consideration in section 2.7 of Materials and Methods (page number 5, lines 193–194).

  • Genes names/abbreviations should be written in italic – this applies also for the Result section

Revised as requested (page numbers 5 and 10; lines 195, 197, 198, 199, 200 and 423, 426, 427, 473).

# Results

  • The results are clearly written, and the figures are overall understandable. There are minor suggestions that could support the description of the findings.
  • Fig. 1 legend – instead of Setting up you could take Experimental design of cell cultures

Revised as requested (lines 366).

  • Fig. 1A – This Figure is helpful and should be enlarged by including the next level of experiments - ACs 5%PL and ACs 10% FCS and CPCs 5% PL

Revised as requested (page number 9). Following a previous Reviewer comment, we highlighted graphically the exclusion of CPC-10% FBS from cell groups examined in this work in order to immediately clarify the absence of this cell control in the analysis.

  • Fig. 3 B – Please include the stained targets Sox9 Col II and Col I in or beside the images in the given color (also Fig. 4B, D; Fig. 6 Col II and Col X). Since Sox9 is a transcription factor and mostly expressed in the cell nucleus – co-illustration with DAPI would be helpful.

Revised as requested (figure 3B, 4C-D, 6A, page numbers 11,12,14).

  • In addition, for the fluorescence images – overviews are quite helpful to support the assumptions (if available).

We apologize for the Reviewer for the non-current availability of the requested image overviews. We acquired the fluorescence images at the reported magnification to better distinguish the cell marker localization. Given the current situation of emergency due to the Covid-19 outbreak we are not able to access in our laboratory and repeat sample acquisitions with fluorescence microscope.

  • Fig. 7 A – Please split the graph in subgroups – otherwise the bars are not readable or comparable; what do the images above stand for? Where are the error bars?

Following the Reviewer comment, we split in half the graph in Figure 7A in the revised manuscript. We showed in the inset above the cytokine array membranes for each conditioned media (CM) tested (CM from ACs-FBS, ACs-PL, and CPCs-PL respectively). In the array membranes, each spot couple represents a molecule (analyte) secreted in the conditioned media. We refer to the manufacturer’s datasheet for the right correspondence between a specific molecule and its position in the membrane of the array kit (R&D Systems, Catalog # ARY022B) as we recalled in the section 2.14 of Materials and Methods (page number 7, lines 316–317). Error bars added in the graph (Figure 7A) of the revised manuscript derived from the calculations on the spot averages.

  • Fig. 7 C – Pease adapt graph to the other graphs – what does fold-increase migration mean? How was normalization performed?

We thank the Reviewer for her/his comment. We proceeded to adapt the graph (Figure 7C) as requested by the Reviewer. The reported “fold increase” migration means that we represented the data by using a frequency scale instead of the absolute values in order to better highlight the observed migration differences compared to the control (migration under the influence of conditioned media from non-treated ACs) set to 1 value). We recalled this specification in section 2.15 in Materials and Methods of the revised manuscript (page number 7, lines 324–327) and in the figure legend (page number 16, lines 611–616).

# Discussion

  • the discussion is really detailed and includes the main findings
  • Line 708 what does CM mean?

Revised as requested with the specification of the abbreviation CM (line 728).

  • Can the authors relate any of the observations to the digestion which was performed for AC isolation but not for CPC?

We thank the Reviewer for her/his comment. We think that the non-digestion of cartilage chips allowed us to preferentially recover some active cells with certain traits. Besides the gain of a migratory ability for the outgrowth from the tissue, the harvested CPCs showed enhanced chondrogenic commitment and an interesting regenerative potential (given by the higher Nestin expression compared to ACs). However, digested cells were a mix of terminally differentiated cells and those cells that were isolated from non-digested tissue. Thus, some of the aforementioned traits appeared also in the latter cell population.

Reviewer 3 Report

The author indicates that the platelet lysate (PL) induces the activation of chondrogenic properties. This is proven using ex vivo treatment. This manuscript is interesting in aspects of therapeutic approach. There are some concerns and they are discussed below.

  1. In the Abstract, the sentence "We describe that the platelet lyset (PL) is able to 26 activate chondroprogenitor cells in a terminally differentiated cartilage tissue" is misprinted. Maybe, the "platelet lyset (PL)" is error-typed (lysate).
  2. Before preparing PL, did the authors check specific cell surface markers for platelet using flow cytometry analysis?
  3. Is there a dose-dependent effect of PL on the proliferation and gene expression of cartilage-derived cells as well as on the production of hyaline-like cartilage in vivo?
  4. In general, description on secretory profile of CPCs and ACs in Result section (3.7) is not easy to follow and is not clear to me. Please shorten the description and clearly describe the results, in terms of the comparison or differences among three groups (CM collected from cultures of ACs-FBS, ACs-PL and CPCs-PL).
  5. In this manuscript, the biopsies were obtained from patient undergoing hip replacement and probably with ongoing degenerative processes. It would be interesting to look at the production of hyaline-like cartilage in vivo using CPCs prepared from the biopsies of normal control people.

Author Response

Reviewer #3

We thank the Reviewer for her/his appreciation of our work. We addressed in depth all her/his comments to further improve quality.

1.In the Abstract, the sentence "We describe that the platelet lyset (PL) is able to 26 activate chondroprogenitor cells in a terminally differentiated cartilage tissue" is misprinted. Maybe, the "platelet lyset (PL)" is error-typed (lysate).

Revised as requested (line 26).

2.Before preparing PL, did the authors check specific cell surface markers for platelet using flow cytometry analysis?

We thank the Reviewer for her/his comment. Platelet-derived products that we adopted for our experiments were scrupulously processed and underwent accurate procedures for which it is not required the check for platelet surface markers in order to ensure standardization. Plasma and buffy layer, that consist of platelet-rich plasma (PRP), was isolated from blood and centrifuged in order to recover platelets. The upper phase, corresponding to the platelet-poor plasma (PPP), was centrifuged once again and this passage repeated at least three times in order to recover the residual platelets suspended. Platelets in the lower phase were automatically counted using an electronic hematology analyzer. Then, the PRP was diluted with PPP at the final concentration of 10 × 106 platelets/µl. Three cycles of freeze-thawing were performed on PRP in order to break the platelet membranes and release the platelet content. After this step, platelet debris and broken membranes were separated by centrifugation at high speed and the supernatant was recovered and stored (see Backly, Tissue Eng Part A., 2011 [27]; Muraglia, Platelets, 2014 [28]). In order to minimize the lot-to-lot variation, a buffy coat pool from several donors was processed for PRP/PL batch preparation. Besides the platelet count that occurs before preparing PL, its components were determined after the described steps to ensure the use in cell cultures, including the content of two representative growth factors: platelet-derived growth factor (PDGF)-BB and vascular endothelial growth factor (VEGF) present at a relatively high and a relatively low level, respectively. Typical results of these assays were reported in the recent work of our laboratory (see Muraglia, Front Bioeng Biotechnol., 2017 [29]). We inserted this reference in section 2.1 of Materials and Methods (page number 3, line 108) to provide further details.

3.Is there a dose-dependent effect of PL on the proliferation and gene expression of cartilage-derived cells as well as on the production of hyaline-like cartilage in vivo?

We thank the Reviewer for the comment which offers us the opportunity to better explain the experimental conditions chosen in the work. In general, the final concentration of PL in cell culture media ranges from 1% to 60%, with 5% to 10% being the most common concentration used. This percentage will depend on the starting platelet concentration, and differs among studies as well as depends on tested cells (see Burnouf, Biomaterials, 2016). In our preliminary experiments, we have already tested several PL concentrations (from 2.5% to 10%)  on chondrocytes and other cell cultures (including mesenchymal stem cells, skin fibroblasts, and osteoblasts). What we have always observed is that 5% PL is the maximum effective concentration in terms of growth stimulation and higher doses have similar effects or even adverse (see Zaky, J Tissue Eng Regen Med., 2008; Muraglia, Platelets, 2014 [28]). We also performed preliminary tests on cartilage chip cultures with a range of increasing PL concentration and even with non-lysed PRP. Once again we recognize 5% PL as the recommended dose to obtain CPC recovery and its reproducibility. Consequently, chondrogenic potential could be tested only in cells obtained in these culture conditions. However, the efficacy of 5% PL dose on cartilage-derived cells was well supported by our previous work (see Pereira, Tissue Eng Part A., 2013 [45]), where it was reported the maintenance of chondrogenic memory in PL-treated chondrocytes both in vitro and in vivo despite the increased proliferation rate (chondrocytes usually lose their phenotype after few replications). Consequently, we think that other concentrations, lower or higher than 5%, can’t allow to value the real contribution of PL in tissue or cell dynamics intended for regeneration. Furthermore, PL added to culture medium at 5% concentration approximately corresponds to the highest physiological platelet dose in human blood, without heparin addition. High concentration gives rise to culture medium gelation, issue resolvable only with supplementation of factors (like heparin) that could change cell responses (see Muraglia, Front Bioeng Biotechnol., 2017 [29]). Data Heterogeneity on dose-related effects of platelet products is common in the scientific literature due to the lack of standardization procedures and consequently changes in growth factor concentrations. We provided this technical information in 2.1 section of Materials and methods (page number 3, lines 109–112).

4.In general, description on secretory profile of CPCs and ACs in Result section (3.7) is not easy to follow and is not clear to me. Please shorten the description and clearly describe the results, in terms of the comparison or differences among three groups (CM collected from cultures of ACs-FBS, ACs-PL and CPCs-PL).

We thank the Reviewer for her/his comment. We adapted the description in section 3.7 of Results (page number 15) in the revised manuscript focusing the attention only on certain aspects that we considered functional to the aim of the study. However, we think that the overview of this analysis providing novel starting points for future detailed studies in the role of CPCs-PL in cartilage regenerative medicine.

5.In this manuscript, the biopsies were obtained from patient undergoing hip replacement and probably with ongoing degenerative processes. It would be interesting to look at the production of hyaline-like cartilage in vivo using CPCs prepared from the biopsies of normal control people.

We thank the Reviewer for her/his suggestion. We think that is an interesting topic to be studied in the next future, although the finding of hip cartilage biopsies from healthy donors is rather rare for us. Furthermore, our supply of bioptic tissue is currently hindered by the Covid-19 outbreak. With this work, we wanted to contribute to understanding PL role in treatment for cartilage disorders and we found the PL-activated CPCs could be a promising tool in the field. Scientific literature has already reported that the chondro-progenitor cell population is scant in normal cartilage tissue and increases in pathological (OA) conditions when probably it involved in repair attempts (see Grogan, Arthritis Res Ther., 2009 [75]; Hattori, Biochem Biophys Res Commun., 2007). However, it was recently demonstrated that mild OA derived cells may have a greater MSC potential than similar severe OA cells, suggesting that differences may exist for chondrogenic potential depending on the severity of OA (see Mazor, Int J Mol Sci., 2017). In light of these findings, our studies on the interaction between platelet derivatives and cartilage could pursue in this direction (i.e. the comparison in PL-treated cartilage from healthy biopsies and gradually degenerated ones) even provide insights for stem-cell based strategies in cartilage treatments and tissue engineering.

Reviewer 4 Report

The manuscript “MSC-like Progenitor Cells Activated by Platelet Lysate in Human Articular Cartilage as a Tool for Future Cartilage Engineering and Reparative Strategies” presents interesting result the manuscript is acceptable for publication after minor corrections.

Comments

The authors should indicate why the chose to use AC-FBS and not CPC-FBS as control.

A table giving the data presented in the section 3.7 (secretory profile) is needed

Data presented on figure 7A should be presented in a table

The discussion is too long and should be shorten to focus on the aim of the study

Author Response

Reviewer #4

The manuscript “MSC-like Progenitor Cells Activated by Platelet Lysate in Human Articular Cartilage as a Tool for Future Cartilage Engineering and Reparative Strategies” presents interesting result the manuscript is acceptable for publication after minor corrections.

We thank the Reviewer for her/his appreciation of our work. We addressed in depth all her/his comments to further improve quality.

Comments

The authors should indicate why the chose to use AC-FBS and not CPC-FBS as control.

We thank the Reviewer for her/his comment. As we reported in the main manuscript (Discussion, page number 17, lines 624–626), the control group CPCs-FBS was not used for comparison with CPCs-PL due to the failed harvesting of cells in cartilage chip culture supplemented with 10% FBS. Although in sporadic cases, we noted some cell outgrowth from tissue fragments, they did not proliferate and could not be expanded in enough quantity and reasonable time to be tested in parallel with the other cell cultures. Consequently, ACs-PL is claimed the control group of CPCs-PL based on the uniformity of the growth conditions (exposure to PL), while ACs-FBS were considered in the comparison with ACs-PL based on the uniformity of isolation method (tissue digestion). We recalled this aspect in the figure 1A (page number 9) of the revised manuscript following the below Reviewer comment and in Discussion (page number 17, lines 624–626 and 633–636).

A table giving the data presented in the section 3.7 (secretory profile) is needed

Following the Reviewer comment, we introduced a table for data derived from cell secretome analysis in Table S1 of Supplementary Materials and recalled in the main text of the revised manuscript (section 3.7 in Results, page number 15, line 538).

Data presented on figure 7A should be presented in a table

Following the Reviewer comment, we integrated the graph reported in Figure 7A with a table in Table S1 of Supplementary Materials (Supplementary Materials indication, page number 20, lines 801–802) in the revised manuscript.

The discussion is too long and should be shorten to focus on the aim of the study

Following the Reviewer comment, we adapted the Discussion in the revised manuscript, focusing on the main key messages that we wanted to communicate.